



# The Impact of Cloud Microphysics and Ice Nucleation on Southern Ocean Clouds Assessed with Single Column Modeling and Instrument Simulators

Andrew Gettelman[1,2], Richard Forbes[3], Roger Marchand[4], Chih-Chieh Chen[1], and Mark Fielding[3]

[1]National Center for Atmospheric Research, Boulder, CO, USA
[2]Pacific Northwest National Laboratory, Richland, WA, USA
[3]European Centre for Medium Range Weather Forecasts, Reading, UK
[4]University of Washington, Seattle, WA, USA

**Correspondence:** A. Gettelman (andrew.gettelman@pnnl.gov)

**Abstract.** Supercooled liquid clouds are common at higher latitudes (especially over the Southern Ocean) and are critical for constraining climate projections. We take advantage of the Macquarie Island Cloud and Radiation Experiment (MICRE) to perform an analysis of observed and simulated cloud processes over the Southern Ocean in a region and season dominated by supercooled liquid clouds. Using a single-column version of the European Centre for Medium-range Weather Forecast's

(ECMWF) Integrated Forecast System (IFS), we compare two different cloud microphysical schemes to ground based observations of cloud, precipitation and radiation over a 2.5 month period (January 1 - March 17, 2017). Both schemes are able to reproduce aspects of the cloud and radiation observations during MICRE to within the uncertainty of the data, when the thermodynamic profile is prescribed with relaxation. There are differences in water mass and representation of reflectivity between the schemes. A sensitivity study of the cloud microphysics schemes, one a bulk one-moment scheme, and the other

a two-moment scheme with prediction of mass and number, indicates that several key processes create differences between the schemes. Surface radiative fluxes and total water path are highly sensitive to the formation and fall speed of precipitation. The prediction of hydrometeor number with the two-moment scheme yields a better comparison with observed reflectivity and radiative fluxes, despite predicting higher liquid water contents than observed. With the two-moment scheme, we are also able to test the sensitivity of the results to the input of liquid Cloud Condensation Nuclei (CCN) and Ice Nuclei (IN). The

cloud properties and resulting radiative effects are found to be sensitive to the CCN and IN concentrations. More CCN and IN increase liquid and ice water paths respectively. Thus both the dynamic environment and aerosols, integrated through the cloud microphysics, are important for properly representing Southern Ocean cloud radiative effects.

## 1 Introduction

Supercooled liquid water, defined as water below the freezing point of 0°C (273K), is common in colder regions of the planet

(Hu et al., 2010). Freezing at temperatures between 0 and -40°C (273 and 233K) typically require very high ice supersaturations and/or Ice Nucleating Particles, also called Ice Nuclei (IN). IN are relatively rare (McCluskey et al., 2018), leading to the presence of supercooled liquid and an important role for secondary ice (number) production processes (Järvinen et al.,



2022). Recent work has made clear that supercooled liquid is important for both weather and climate. For example, improving supercooled liquid has been shown to improve forecasts of 2m temperatures at high-latitudes over land (Forbes and Ahlgrimm, 2014) and cold air outbreaks at high latitudes (Forbes et al., 2015), while Bodas-Salcedo et al. (2012) showed a deficiency of supercooled liquid in several climate models and demonstrated how this deficiency resulted in substantial errors in the radiative budget of the Southern Ocean. More generally, Bodas-Salcedo et al. (2019) and Gettelman et al. (2019b) illustrated in two different models how supercooled liquid clouds can impact cloud feedbacks and the climate sensitivity of the planet to an imposed forcing.

There is a longstanding bias in climate (Trenberth and Fasullo, 2010) and weather forecast (Forbes et al., 2015) models of too few Southern Ocean clouds and too much surface absorption of solar radiation. To reduce cloud model biases at mid- to high-latitudes, it is necessary to focus on the representation of cloud phase and precipitation, which is handled by the cloud microphysical scheme in a large scale model. In this manuscript we present observational and model comparisons over a season for supercooled liquid clouds over the Southern Ocean. We attempt to reproduce the observations in a single column version of the ECMWF IFS (Integrated Forecast System) with cloud microphysics schemes from both a weather forecast model (the ECMWF IFS) and a climate model (Community Earth System Model version 2, CESM2).

The unique aspects to this study are several. First, we compare two 'operational' cloud microphysics codes in a region with complex cloud regimes within the same modelling framework. Forbes and Tompkins (2011) describe a bulk, one-moment (predicted mass only) scheme designed for weather forecasting. The other scheme (Gettelman et al., 2019a) is a two-moment (predicted mass and number) scheme designed for climate simulation. We seek to understand what is similar between the schemes, what is different and how the differences affect model biases relative to the observations. We use a constrained model set up, with a single column model forced by observed winds and temperatures over a 75 day austral summer period. This allows a statistical representation of a series of events and regimes, while also being able to look at detailed process rates and a close comparison to observations. Finally, we focus on clouds with extensive supercooled liquid, a long-standing issue for weather and climate models.

Section 2 discusses the simulation set up, the microphysical parametrization schemes, and the observations. Section 3 presents the results, including comparisons to the observations, and comparisons between the two schemes. We also present sensitivity studies of the model configurations to understand the role of different microphysical processes and why the schemes differ. A summary and conclusions are in Section 4.

## 2 Methodology

Here we first describe the observations (Section 2.1), followed by the IFS Single Column Model, the IFS microphysical and the MG3 microphysical schemes, the model radar simulator in (Section 2.2), and details of the simulations (Section 2.3).



## 2.1 MICRE Observations

During the Macquarie Island Cloud and Radiation Experiment (MICRE), observations were collected from March 2016 to March 2018 at Macquarie Island (54.5° S, 158.9°E), located about half-way between New Zealand and Antarctica. The project is described in detail in McFarquhar et al. (2020), and included deployment of a suite of ground-based instruments including a ceilometer, surface rain disdrometer, microwave radiometer, and broadband shortwave (SW) and longwave (LW) radiometers, as well as an upward looking cloud radar and depolarization lidar. Most of the instruments suffered periods with significant down time, and we focus on the 75-day period from January 1 to March 17, 2017, when upward looking cloud radar and microwave radiometer observations are available, in addition to twice-daily radiosondes and broadband radiometer fluxes. We focus on observations of surface downward radiation, radar reflectivity, and microwave-radiometer-derived liquid water path, as well as the ancillary radiosonde temperature data. The MICRE broadband radiometer SW and LW fluxes have been evaluated and compared with Clouds and the Earth's Radiant Energy System (CERES) satellite synoptic (SYN) product (Doelling et al., 2013; Rutan et al., 2015) and the surface Energy Balanced and Filled (surface EBAF, Kato et al., 2018) fluxes by Hinkelman and Marchand (2020).

## 2.2 Model Description

In this study we run a version of the Integrated Forecast System (IFS) (ECMWF, 2019) in single column mode with two different cloud microphysical schemes. Both are 'bulk' schemes representing hydrometeors with a mean mass, and one is a two-moment scheme that also represents the mean number of hydrometeors. The schemes however often use similar bulk formulations for process rates, either solely mass based, or with an assumed size in the one-moment (mass only) scheme. This makes for an interesting and direct set of process rate comparisons, and highlights where two-moment schemes may be necessary to capture different atmospheric process sensitivities and regimes.

### 2.2.1 IFS Single Column Model

The Single Column Model (SCM) is a standalone vertical column version of the IFS with the same suite of physical parameterizations as the global model. Time-varying surface and advective forcings can be specified as well as a relaxation towards a specified time-varying state with a given timescale. The particular version used here is IFS Cycle 46r1, used operationally at ECMWF from June 2019 to June 2020. There are 137 vertical levels, the same as the operational global model, with a layer depth increasing from 20m to 170m in the lowest 2km and around 300m in the rest of the troposphere. Only an overview of the parameterizations is given here and further details can be found in the IFS 46r1 documentation (ECMWF, 2019).

The IFS has prognostic variables for cloud fraction, specific humidity, specific cloud liquid, cloud ice, rain and snow water contents. Saturation adjustment is part of the sub-grid cloud parameterization scheme, based on Tiedtke (1993) with sources and sinks from the vertical advection, radiation and convection parameterizations. Supersaturation with respect to ice is allowed and the assumptions are described in Tompkins et al. (2007). The cloud scheme is tightly coupled with the convection parameterization with detrainment of cloud fraction, condensate and precipitation from sub-grid convective updraughts. Verti-





85 cal advection due to convectively generated sub-grid subsidence within a convecting grid cell is represented, as well as turbulent erosion of cloud fraction and condensate at sub-grid cloud edges. The cloud and precipitation microphysics is described in the next subsection.

The parameterization of shallow, mid-level and deep convection is based on the mass-flux approach (Tiedtke, 1989; Bechtold et al., 2008, 2014). The turbulent mixing scheme follows the Eddy-Diffusivity Mass-Flux (EDMF) framework, with a K-

90 diffusion turbulence closure and a mass-flux component to represent the non-local eddy fluxes in unstable boundary layers (Siebesma et al., 2007; Köhler et al., 2011). The radiation scheme (ecRad) is described in Hogan and Bozzo (2018) with the gas optics from the Rapid Radiation Transfer Model (RRTMG, Mlawer et al., 1997; Iacono et al., 2008). Cloud-radiation interactions are taken into account using the McICA (Monte Carlo Independent Column Approximation) method (Morcrette et al., 2008). Surface exchange and gravity wave drag are also represented (Balsamo et al., 2009; Lott and Miller, 1997; Beljaars

et al., 2004; Orr et al., 2010).

### 2.2.2 IFS Microphysics

The IFS microphysics is a one-moment bulk scheme with prognostic variables for the mass of 4 classes of hydrometeor (liquid, ice, rain, snow) as described in Forbes and Tompkins (2011) and Forbes et al. (2011) with various modifications, particularly to improve the representation of mixed-phase boundary layer clouds (Forbes and Ahlgrimm, 2014) and warm-rain processes

(Ahlgrimm and Forbes, 2014). The numerical formulation of the hydrometeor sedimentation follows an implicit upstream approach with parameterized fall speed for rain (Sachidananda and Zrnic, 1986; Abel and Boutle, 2012) and fixed fallspeeds for snow (1 m/s) and ice (0.13 m/s).

### 2.2.3 MG3 Microphysics

The Morrison-Gettelman microphysics scheme version 3 (MG3) is a two-moment bulk microphysics scheme with prognostic

variables for mass and number concentration of 5 classes of hydrometeors (liquid, ice, rain, snow, graupel) as described by Gettelman et al. (2019a). MG3 is based on the original Morrison et al. (2005) scheme, adapted and modified extensively for climate models (Morrison and Gettelman, 2008), with extensions for ice nucleation (Gettelman et al., 2010) and prognostic precipitation (Gettelman and Morrison, 2015). The scheme without graupel is used in the Community Earth System Model version 2 (CESM2, Danabasoglu et al., 2020). The scheme compares well to mesoscale schemes by Morrison et al. (2005) and

Thompson and Eidhammer (2014) for idealized shallow and convective cloud cases, as shown by Gettelman et al. (2019a).

For this study MG3 has been adapted for use in the IFS SCM as an alternative to the IFS parameterization of microphysical processes and sedimentation. The rest of the model is kept the same including the saturation adjustment, sub-grid cloud parameterization and convection interactions.

The additional MG3 prognostic variables for mass and number concentration of all 5 classes of hydrometeors have been

added to the IFS SCM. The two-moment MG3 scheme can run with fixed hydrometeor number concentration, or use activated number concentration rates ($\# \text{ s}^{-1}$) for liquid drops (cloud condensation nuclei, CCN) and ice crystals (ice nuclei, IN). Since the IFS does not have an explicit representation of aerosols or an activation scheme, for this work we assume constant CCN



and IN activation rates of 1 cm$^{-3}$ s$^{-1}$ and 5L$^{-1}$ s$^{-1}$ respectively for all altitudes at every timestep. These rates are just used to initialize new particles in the microphysics when the model physics condenses water or ice and we test the sensitivity of the results to the activation/nucleation rate.

### 2.2.4  Radar Simulator

The radar simulator, described in full in Fielding and Janisková (2020), provides the model-equivalent radar reflectivity factor to compare with observations. It has been adapted to run in-line within the SCM and, wherever possible, uses assumptions consistent with the two microphysics schemes used in this study. The simulator runs in 'one-moment' mode when the IFS microphysics scheme is used and 'two-moment' mode when the MG3 scheme is used. When in two-moment mode, the simulator computes bulk scattering properties by combining the MG3 particle size distributions with the prognostic number concentration, mass and the same single scattering properties as the one-moment simulator.

In-cloud hydrometeor masses (and prognostic number concentrations when in 'two-moment' mode) for the different species are converted to an unattenuated radar reflectivity via the drop size distributions assumed by the microphysics schemes. The simulator then finds the attenuated radar reflectivity by calculating the transmission between each model layer while accounting for attenuation from both gases and hydrometeors and sub-grid cloud overlap.

### 2.3  Simulation Setup

The simulations are set up by generating forcing files for the location of Macquarie Island station (54.5°S, 158.9°E) from January to March 2017. The forcing comes from CESM2 simulations relaxed to the MERRA2 reanalysis (Molod et al., 2015), similar to that used by Gettelman et al. (2020) for the Southern Ocean Clouds, Radiation, Aerosol, Transport Experimental Study (SOCRATES) project. The IFS SCM is relaxed back to the forcing profile with an 8 hour relaxation time for winds, temperature and humidity to keep the boundary layer structure from drifting too far from the radiosonde observations. Longer relaxation timescales, or no relaxation at all, caused the simulations to drift by several °C in the upper part of the boundary layer, resulting in significant biases in clouds relative to the observations. This is a key feature: getting boundary layer structure correct is vital for constraining cloud radiative effects. There is no relaxation of cloud variables and these are allowed to freely evolve during the simulation. The base time step for the simulations is 225s, but we also investigate the time step sensitivity of both schemes by running at 60s and 900s (the time step used in the global forecast model at ECMWF depends on resolution but typically ranges from 225s to 900s).

Simulations are repeated with the IFS and MG3 microphysics for comparison, and a series of sensitivity tests are performed, modifying different aspects of the MG3 code (see section 3.2.3). The CESM2 version of the MG3 microphysics was initially implemented in the SCM, but first comparisons highlighted significant differences in the profiles of rain that were due to differences in a few basic assumptions in the IFS. To allow a more meaningful comparison of the process rates, it was decided to implement these IFS assumptions in the MG3 scheme. Rain evaporation is modified to more closely match the IFS, with a scaling factor of 0.3 and a relative humidity threshold between 80% and 90% before evaporation can occur, both currently required in the IFS to reduce excessive evaporation in the operational IFS. The threshold for rain freezing is changed from



-40°C in CESM2 to the value of -5°C in the IFS, so all precipitation below -5°C is treated as snow. In addition, a correction for fall speeds of hydrometeors is added to prevent the formation of layers with zero fall velocity (which particularly affects rain), and the sedimentation of precipitation is changed to an implicit monotonic scheme (Harris et al., 2020; Zhou et al., 2019) to reduce timestep sensitivity. Finally, there is one change for the ice microphysics: mixed-phase ice nuclei are determined
by the Meyers et al. (1992) empirical function of temperature as in Morrison et al. (2005) and the IFS, rather than the Hoose et al. (2010) classical nucleation theory scheme used in CESM2. These changes together define the 'base' version of the MG3 scheme used for the comparison with the IFS in section 3, although the original CESM2 version of the MG3 scheme is also included for comparison in the sensitivity study (section 3.2.3).

## 3   Results

First we compare the IFS and MG3 microphysics (modified from the CESM2 version as described in section 2.3) to observations from MICRE. We then examine the differences between the IFS and MG3 microphysics, including their timestep sensitivity, differences in the balance of process rates between the schemes, and finally sensitivity tests with the MG3 scheme.

### 3.1   Comparison to Observations

In this subsection, we compare model simulations to surface and satellite based observations. We begin by focusing on short-
wave and longwave radiative fluxes, followed by in-cloud liquid water path (LWP) and radar reflectivity (obtained from the model via a radar simulator as described in section 2).

### 3.1.1   Top-of-Atmosphere and Surface Radiation

Figure 1 shows the downward surface shortwave (SW) and longwave (LW) radiative fluxes for the 75-day time series, for the surface radiometer (SFC, black line), the CERES SYN product retrieved from combined geostationary and MODIS satellite
data (for the SW flux, SYN, blue), and the simulation results from the MG3 (orange) and IFS (green) microphysical schemes. Although there are short periods during which the model simulated fluxes depart noticeably from the surface and satellite data, there is generally good agreement. Hourly correlations for both simulations with the observations are ∼0.75 for the SW flux and ∼0.6 for the LW, and the mean SW flux for both simulations lies between the mean values from the surface and satellite data sets. The one-sigma sampling uncertainty in the surface and satellite daily mean SW flux is about 17 $Wm^{-2}$ and 5 $Wm^{-2}$
for the LW. The sampling uncertainty is estimated as the standard deviation divided by the square root of the number of days, effectively treating each day as equivalent to a single independent sample. The difference between the surface and satellite mean SW flux might be due to a bias in one or both data sets, or due to differences in the field of view (for example, it is possible that it is slightly cloudier or clouds are less optically transparent over the island site than the nearby ocean) but we do not know for certain the source of the mean difference. In short, the small differences between all of the mean SW fluxes shown
in top panel of Figure 1 are not significant with respect to the sampling uncertainty. Sampling and systematic uncertainties in the surface and satellite fluxes are discussed in more detail by Hinkelman and Marchand (2020). Figure 1 also shows the surface



downward longwave (middle panel) with a strong correspondence between the simulated and observed fluxes. Here the SYN retrievals are not shown because there is a significant bias in the CERES SYN retrievals at night (Hinkelman and Marchand, 2020). The small differences (<5 Wm$^{-2}$) between the mean model LW fluxes and the observed mean value are likewise not

significant with respect to the sampling uncertainty.

We examine the diurnal cycle of surface and top of atmosphere (TOA) SW and LW fluxes in Figure 2. The top two panels in the left column show the observed and simulated diurnal cycle of outgoing TOA SW and LW flux, followed below by the downward surface SW and LW flux, and the mean in-cloud LWP (bottom panel). We will return to the LWP momentarily. The panels in the right columns are the same but restricted to periods with low clouds, defined as times when hydrometeors are

present at or below 2 km but not above 2 km, based on the observed or simulated hourly-averaged radar reflectivity. Each hour in the surface (MICRE) and CERES SYN datasets is defined as low cloud based on the surface radar observations. For both the models and the observational datasets, hours containing only low clouds occurred about 35% of the time and completely clear hours only about 2%, with the remainder containing some hydrometeors above 2 km (most often with hydrometer both above and below 2km). In Figure 2 each dataset is processed independently, and we have not restricted the low cloud times

to those hours in which only low cloud occur at the same time in both surface observations and model simulations. Doing so significantly decreases the total number of hours and increases the estimated sampling uncertainties, but does not otherwise qualitatively change the results. We examine the observed and simulated radar reflectivity and associated vertical profiles of cloud occurrence in more detail later in this section.

The simulated diurnal cycle of surface downward SW and LW fluxes compare well with the surface and satellite data; and

the good agreement holds for low cloud periods. There is no discernible diurnal cycle in the observed or modeled surface LW flux. The mean surface SW downward flux is about 30 Wm$^{-2}$ larger during low cloud periods as compared with all times, while the LW downward flux is about 8 Wm$^{-2}$ smaller. This demonstrates the significant role played by both shallow and deeper clouds in the radiative energy budget, and the models capture this well. The modeled outgoing TOA flux also agrees reasonably well with the CERES SYN product. The model (daily) means are within about 15 Wm$^{-2}$ of the SYN mean, with

the CERES SYN product being a bit smaller (consistent with the downward CERES SYN surface SW flux being a bit larger than the model simulated surface SW flux). For clarity, the sampling uncertainty is not shown on the model simulated curves. If displayed, the one-sigma uncertainty in the model mean diurnal cycle would just overlap with the uncertainty shown for the observations. A paired data test (not shown) indicates the 15 Wm$^{-2}$ difference between the models and SYN data is significant at the one-sigma level. Nonetheless, it remains a reasonable possibility that the 15 Wm$^{-2}$ difference is just a result of sampling

uncertainty. The same is not true for the outgoing TOA LW flux. In the LW, while there is good agreement between the model simulated outgoing TOA LW and the CERES SYN product during periods with low clouds for both simulations (right panel), under all sky conditions the IFS microphysical scheme has too little outgoing LW flux (and this is significant at the two-sigma level in a paired data test). The underestimate in the IFS flux becomes even clearer when restricted to hours where only high-level hydrometeors are present (not shown). As will be shown later, an examination of radar reflectivity fields likewise reveals

the simulations with the IFS microphysics have a higher occurrence of cloud above 6 km and we will return to this topic after an examination of the liquid water path.





**Figure 1.** Time series of hourly averaged values: (top panel) downward surface shortwave (SW) flux, (middle panel) downward longwave (LW) flux, and (bottom panel) liquid water path (LWP). For each dataset the legend gives the mean value, and for the SW and LW flux, the estimated one-sigma uncertainty in the mean is also given. For LWP, the legend gives the mean value for all times followed by the mean restricted to periods where only low clouds are present (see text). The CERES SYN data and average values are also restricted to data collected between 9 and 16 hours local time to avoid biases associated with low solar zenith angles (see text).



**Figure 2.** Mean diurnal cycles. Panels from top to bottom show the outgoing TOA SW and LW flux, downward surface SW and LW flux, and LWP. Left column shows diurnal averages for all times and the right column for periods when only low cloud (hydrometeors < 2 km) are present. In each panel, shading that surrounds the surface or satellite values depicts the estimated one-sigma sampling uncertainty for a particular hour over all days. Sampling uncertainty for all datasets is of similar magnitude. The bottom panel in each column shows the diurnal cycle of LWP. CERES SYN retrieved LWP in purple (which combines geostationary and MODIS data). MODIS only (red symbols) as described in the text. The legend gives the daily mean for each dataset, and for LWP both the daily mean and median are given. CERES SYN means and medians are restricted to data collected between 9 and 16 hours local time (see text).



### 3.1.2 Liquid Water Path

The timeseries of LWP is shown in the bottom panel of Figure 1, and the mean diurnal cycle for all and low cloud conditions is shown in bottom row of Figure 2. Both figures depict the mean in-cloud LWP, with the model in-cloud value taken as the grid

mean value divided by the model prognostic cloud fraction. Retrievals of the LWP based on the surface microwave radiometer (MWR) are only accurate when the microwave radiometer is dry (when it has not recently rained or even drizzled heavily), and so the surface (SFC) MWR data include only periods when low-and-non-precipitating cloud are present. The SFC LWP was derived using a physical-iterative approach (Marchand et al., 2003), with a structural uncertainty of about 15 to 30 $gm^{-2}$ (owing primarily to uncertainties in the underlying microwave spectroscopy and the treatment of supercooled liquid water

absorption). Here the structural uncertainty is larger than the sampling uncertainty. On a minor note, the very small differences between the mean LWP values given in the legends in Figure 1 and Figure 2 arise because in Figure 2 the data are first averaged in each hour before the overall (daily) mean is taken and the number of samples is not constant throughout the day, but depends on when low cloud conditions occur.

The CERES SYN mean (and median) values given in the legend of both figures are restricted to data collected between 9 and

16 hours local time. The diurnal cycle plots show there are large nonphysical "spikes" in the CERES SYN LWP before and after this time, approaching sunrise and sunset. During daylight, the CERES SYN cloud property retrieval is based on visible and near infrared imagery and assumes that clouds (whether they are liquid or ice phase) are sufficiently spatially homogeneous that scattering at these wavelengths can be treated using a one-dimensional (plane-parallel) approximation. Details on the CERES retrieval methodology are given in Minnis et al. (2011, 2008) and largely follow the approach used in CERES-MODIS

products. The plane-parallel approximation is widely known to work poorly at large solar zenith angles or large view angles, and we speculate that these spikes are a result of this approximation not working well. The CERES SYN product does use an infrared-only algorithm at night, and we also speculate that the lower SYN LWP values observed during nighttime are also a retrieval artifact (and may play a role in the nighttime downward LW surface bias in the SYN product noted by Hinkelman and Marchand (2020)). As a point of comparison, we have also plotted the mean plus/minus one-standard-error in the operational

MODIS MOD06 and MYD06 mean LWP (Platnick et al., 2003, 2017), shown by the red bars (different than the CERES MODIS algorithm). While MODIS, on board the Terra and Aqua satellites, orbits in a sun-synchronous orbit, Macquarie Island is far enough south that it is often observed on the edges of the MODIS swath resulting in local overpasses that can be somewhat earlier or later than the nominal 10:30 and 13:30 equator crossing times. The mean MOD06 and MYD06 LWP, which are processed independently of the CERES project, are in reasonable agreement with the CERES SYN data.

As is apparent in Figure 2, the MG3 microphysics results in a LWP that is too large on average, even when restricted to periods of low clouds. The time series of LWP in Figure 1 shows that the MG3 scheme results in much larger LWP during periods when the IFS scheme is relatively large (more than 300 g $m^{-2}$, e.g. near 350 and 1200 hours since January 1), while at other times it is noticeably lower than IFS and the observational data (e.g. near 250, 800 and 1000 hours since January 1). How is it that the MG3 shortwave radiative fluxes show little bias with respect to observations while there is a large bias in LWP?

Part of the answer to this apparent contradiction is that the distribution of LWP is highly skewed, where a small occurrence of





large LWP events substantially increases the mean. The median values of MG3 LWP are comparable to those for the IFS and the observational datasets (the mean / median values are given in the legend of Figure 2). The nonlinear relationship between albedo and LWP is also a factor. Figure 3 plots the TOA SW albedo as a function of the LWP for the low cloud periods. The MG3 simulation clearly has more high LWP events, but the high LWP does not substantially increase the SW albedo because

the albedo begins to saturate when the LWP is much above about 300 g m$^{-2}$ with little change in albedo (from 0.55-0.65) for large changes in LWP. The partially-offsetting greater occurrence of low LWP events in MG3 is also evident in Figure 3. We note that ice condensate also has a major effect on the SW albedo in deeper clouds and likely reduces the impact of the large LWP events on albedo, but the cloud condensate below 2 km is predominantly liquid.

### 3.1.3 Radar Reflectivity

Figure 4 illustrates the observed (top) and simulated reflectivity from the simulations with MG3 (middle) and IFS (bottom) microphysics. Reflectivity is estimated using a radar simulator as described in Section 2. Liquid, ice, rain and snow and their attenuation are included. In this figure, the observed and simulated reflectivity are averaged over an hour (in power or Z-space), with the hourly values given here as equivalent radar reflectivity dBZe (where dBZe=$10log_{10}(Ze)$). As with the SW and LW fluxes, there is good correspondence between the simulations and observations with both depicting synoptic events with deep

cloud layers that have high reflectivity, and periods with persistent low cloud. However, compared to the observations the models show a greater occurrence of high reflectivity near the surface related to rain (red colors), especially beneath the deep clouds, and a greater occurrence of low reflectivity (light blue colors) in the upper troposphere related to ice cloud.

To better understand these differences, we look at reflectivity-height histograms. The upper left panel of Figure 5 shows the reflectivity-height histogram for the surface (vertically pointing W-band) radar, followed below by simulated histograms

using the MG3 (middle left) and IFS (bottom left) microphysics. The reflectivity data shown here are not hourly-averages. Rather the observational surface radar data has a temporal resolution of about 12 seconds (and so represents a horizontal grid scale less than one km), while the radar simulator reflectivities depend on the assumed model sub-grid distribution (more on this below). The surface radar histogram has very few detections to the left of the minimum detectable signal (MDS) of the radar (violet line). For reference, the MDS is also plotted on the model simulated histograms. The model simulations

contain a large occurrence of hydrometeors above 5 km that the surface radar would be unable to detect. To a degree, this detection limit explains the larger occurrence of low-reflectivity hydrometeors in the model simulations (noted in connection with the time series in Figure 4). The IFS microphysics produces more of this low-reflectivity, high-altitude cloud than the MG3 microphysics. While the surface radar does not help to evaluate the models in this regard, we note that earlier it was found that the IFS microphysics produces too little outgoing LW flux (during periods with deeper clouds) while the MG3 microphysics

compared well to the CERES SYN product. This suggests that the MG3 is likely better representing this high-altitude cloud.

Above the surface radar MDS (to the right of the violet line), both simulations produce a higher occurrence of hydrometeors than observed (note the many orange and red colors in the simulated reflectivity-height histograms that are absent in the observations). This is also demonstrated in the top right panel which displays the vertical profile of hydrometeor occurrence counting only detections above the radar MDS. We stress that the occurrence of hydrometeors above the MDS is strongly





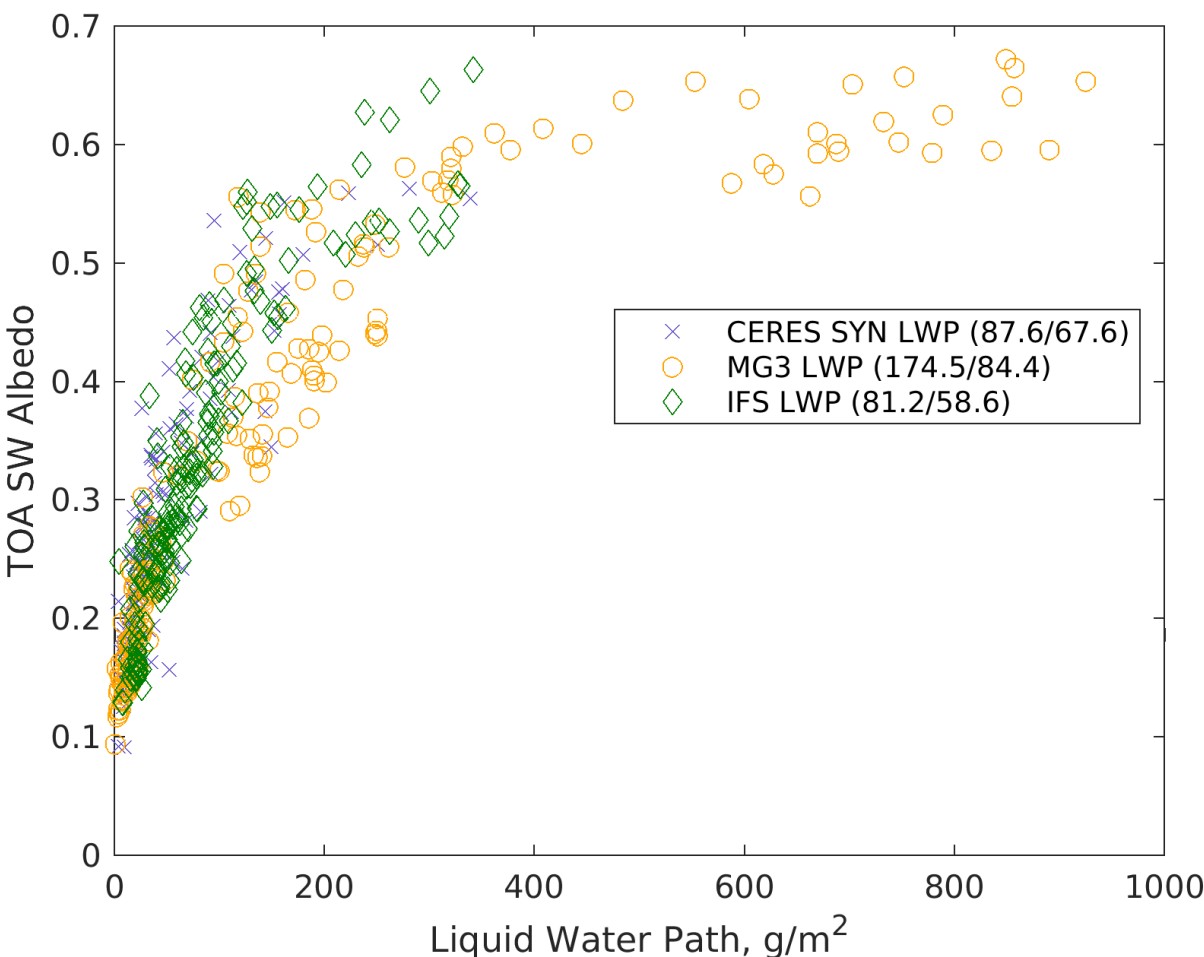

**Figure 3.** TOA SW albedo from CERES SYN as a function of in-cloud liquid water path (LWP) for low-cloud periods during daylight (local time from 9 to 16 hours) from CERES Observations (purple cross), MG3 simulation (orange circles) and IFS simulation (green diamond). Legend shows the mean/median LWP.





**Figure 4.** Hourly mean reflectivity from the observations (Top) and SCM simulations with MG3 (Middle) and IFS microphysics (Bottom).





affected by the model sub-grid representation. Specifically, in the model radar simulator, clouds and precipitation are assumed to be uniformly distributed in the horizontal (e.g., liquid and ice water content do not vary horizontally) and to have the same fractional coverage when both exists (meaning the precipitation fraction equals cloud fraction). The cloud fraction is given by the largest of (i) the model prognostic cloud fraction, (ii) the stratiform precipitation fraction, or (iii) a fixed value of 10% representing convective precipitation. So for example, if the model prognostic cloud fraction is 78% and convective

precipitation is present (at a given vertical level) then both the cloud and precipitation are assumed to cover 78% of the region (at the given vertical level) and the amount of "in-cloud" condensate will be equal to the grid-mean amount divided by 0.78. This uniform horizontal distribution has a significant affect on the simulated radar reflectivities. An analysis by Hillman et al. (2018) found these approximations cause the simulated radar reflectivity histograms to narrow around a "characteristic curve" where the reflectivity associated with precipitation most frequently occurs. In the top left panel of Figure 5, the characteristic

curve for the observations is depicted by the solid black line. The effect of the model assumption is especially evident near the surface, below about 2 km. Here the surface radar shows two distinct modes, a low reflectivity mode with a peak near -25 dBZe and a high reflectivity mode with a peak near 0 dBZe. This is also clear in the bottom right panel (black line), which plots the mean observed reflectivity distribution below 2 km for both the observations and the simulations. The presence of these two modes is typical of most regions (Marchand et al., 2009) and is associated with non-precipitating (low reflectivity) cloud on

the left and precipitation (high reflectivity) on the right. On a minor note, these modes are much less distinct over the Southern Ocean in the spring through fall, when there is more frequently drizzle, as well as frozen (snow) and mixed-phase precipitation with intermediate reflectivity factors (not shown). The simulations (especially with the IFS microphysics) show very little low reflectivity cloud near the surface. This is because some precipitation is almost always present in the simulations and even a small quantity of precipitation can generate a significant radar reflectivity. Thus while the model simulated total hydrometeor

occurrence near the surface is only slightly larger than observed (upper right panel), the assumption that precipitation (when present) is coincident with cloud results in a large overestimate in the occurrence of reflectivity factors larger than about -20 dBZe and an underestimate below -20 dBZe.

Of course, one could make a completely different assumption about the horizontal distribution of condensate and about the co-occurrence of cloud and precipitation (i.e. not maximize the horizontal cloud and precipitation overlap) in the radar

simulator, and thereby obtain a simulated histogram that more closely matches the observations. But doing so only in the context of the simulator and without accounting for this sub-grid variability in a consistent way throughout the microphysics scheme is problematic. In our view, the point of using a radar simulator in the present analysis is to help evaluate the model physics, rather than trying to predict what a radar might observe. While in some sense the conclusion that cloud-and-precipitation shouldn't be maximally overlapped is obvious; on the positive side, the analysis shows the total low cloud cover in the simulations

is reasonable and also points to the potential value of these radar data in helping develop more sophisticated microphysics schemes that can represent sub-grid variability (and such schemes would nominally be better able to account for cloud-aerosol-precipitation interactions).

Returning to the topic of the characteristic curve, Hillman et al. (2018) suggest that while the distribution of reflectivities around the curve is sensitive to the sub-grid distribution, the position of the characteristic curve is not very sensitive to this



representation. For comparison purposes the observed characteristic curve (shown by the solid black line) is plotted on top of the simulated histograms, with the simulated characteristic curves given by the solid white lines. With the IFS microphysics (bottom left panel), there is good agreement between the simulated and observed characteristic curves between 1 and 5 km, suggesting that typical modeled precipitating ice water content (and the underlying parameterized/fixed particle size in the one-moment scheme) are quite reasonable. With the MG3 microphysics, on the other hand, the characteristic curve is shifted

left between 1 and perhaps 4 km (and is located at notable smaller reflectivity factors). As we will see in the next section, the mean ice water content in the MG3 and IFS experiments are similar, and the difference in the characteristic curve is likely due in some combination to the additional attenuation caused by having too much liquid water in MG3 or the particle size (which is not fixed) being too small. In both experiments, below 1 km the characteristic reflectivity is low (left of the observations) and there is a large occurrence of reflectivity values between -15 and -5 dBZe relative to the observations. This is consisting with

the early discussion of precipitation being spread over too wide an area (too frequent and consequently too light). Above 5 km, the MDS is likely affecting the observed characteristic curve in the observations and shouldn't be compared directly with the model, but as noted above, IFS clearly has larger volume of ice and a higher occurrence of smaller reflectivity factors.

## 3.2 Model Simulation Differences

In this section we explore the differences between the MG3 and IFS microphysics schemes in more detail.

### 3.2.1 Mean Hydrometeor Profile and Timestep Sensitivity

Figure 6 shows the mean profiles of key hydrometeors for the IFS and MG3 microphysical schemes. There is overall more supercooled liquid water (SLW, Figure 6A) in the MG3 scheme than the IFS, consistent with the LWP discussion in Section 3.1.2, with almost a factor of two more at altitudes below 750hPa, but less SLW above. Total ice (ice, snow and graupel, Figure 6B) is quite similar between the two schemes, as is the total precipitation (rain, snow and graupel, Figure 6C). A common problem

is a sensitivity of microphysical processes to the length of the timestep, particularly for sedimentation, with various different numerical solutions proposed with different accuracy and stability. Figure 6 also shows the sensitivity of the mean profile of supercooled liquid water, total ice and precipitation, as well as liquid and ice number concentration (for MG3) to the timestep used in the MG3 and IFS microphysics schemes. A range of timesteps, specifically 60, 225 and 900s are shown for each microphysics scheme. The default used in the base line simulations described above is 225s. The IFS microphysics is insensitive

to timestep, due to the fact that it is has all the microphysical process calculations inside an implicit sedimentation loop for each model level. As the CESM2 version of the MG3 scheme has a significant timestep sensitivity (not shown), a modification was made to change the sedimentation numerics from an explicit to an implicit calculation (as discussed in section 3.2.3). It is the modified MG3 scheme that is used here and Fig. 6 shows it is quite stable with respect to timestep, with some differences in peak supercooled liquid water (SLW) (Figure 6A) and in ice number (Figure 6E), particularly for the long (900s) timestep.







**Figure 5.** Left panels: Radar reflectivity-height histograms from surface (vertically pointing W-band) radar, and from simulations using a radar-simulator with the MG3 and IFS microphysics. Colors indicate the fraction of time that hydrometeors are detected at the specified altitude and reflectivity. Histogram bins are 0.5 km x 5 dBZe. Violet line shows the Mininum Detectatable Signal (MDS) of the surface radar. Black and white lines show the characteristic curve (see text) for observations (black, same in all panels) and simulations (white). Top-right: profile of total hydrometeor occurrence, including only detections above the MDS (violet line). Middle-right: Mean distribution of radar reflectivity for hydrometeors above 2 km altitudes (note occurrence in OBS is strongly affected by MDS below about -20 dBZe) . Bottom-right: Same as above but includes only hydrometeors below 2 km (occurrence in OBS is unaffected by MDS above about -35 dBZe)



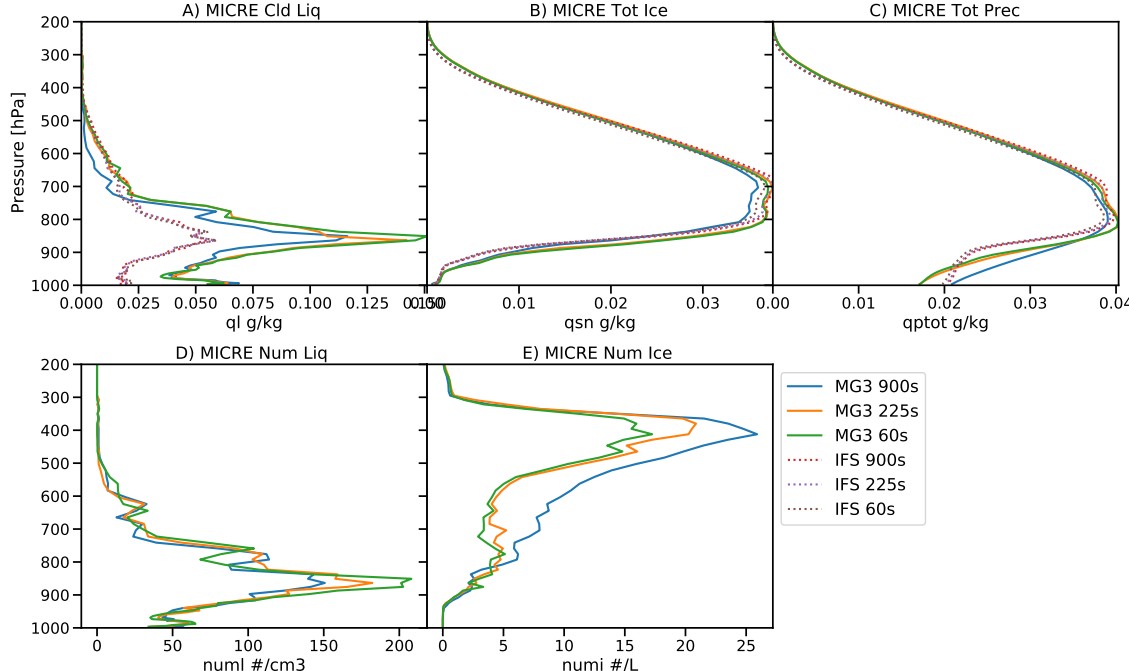

**Figure 6.** Vertical time averages of A) Supercooled Liquid Water, B) Total Ice (ice+snow+graupel), C) Total Precipitation, D) Liquid Number Concentration and E) Ice Number concentration, for Reference IFS simulations (dotted) and MG3 microphysics simulations (solid) at different time steps.

### 3.2.2 Microphysical Process Rates

The two schemes have many similarities but also significant differences in the formulation of the microphysical process rates, and in this section we explore the balance of individual process rates for each hydrometeor (cloud liquid, ice, rain and snow) in both schemes. We do not break this down by regime, but look at process rates for all conditions.

Figure 7 illustrates different process rates that sum to give the total sink (loss) tendency for cloud liquid water. These are the microphysical process rates after the cloud scheme has produced large scale condensation by removing supersaturation (i.e. the source term for cloud condensate which therefore has the same formulation for both simulations). MG3 represents more microphysical processes than the IFS due primarily to the more detailed representation of ice nucleation and the addition of graupel. The mixed phase vapor deposition (or Wegener-Bergeron-Findeisen or 'Bergeron') process for the growth of ice particles is the most important loss process for supercooled liquid water droplets at upper levels in both schemes. At lower altitudes the collision (accretion) of snow with liquid droplets ('Accre snow') dominates and rain collision-coalesence ('Accretion') below





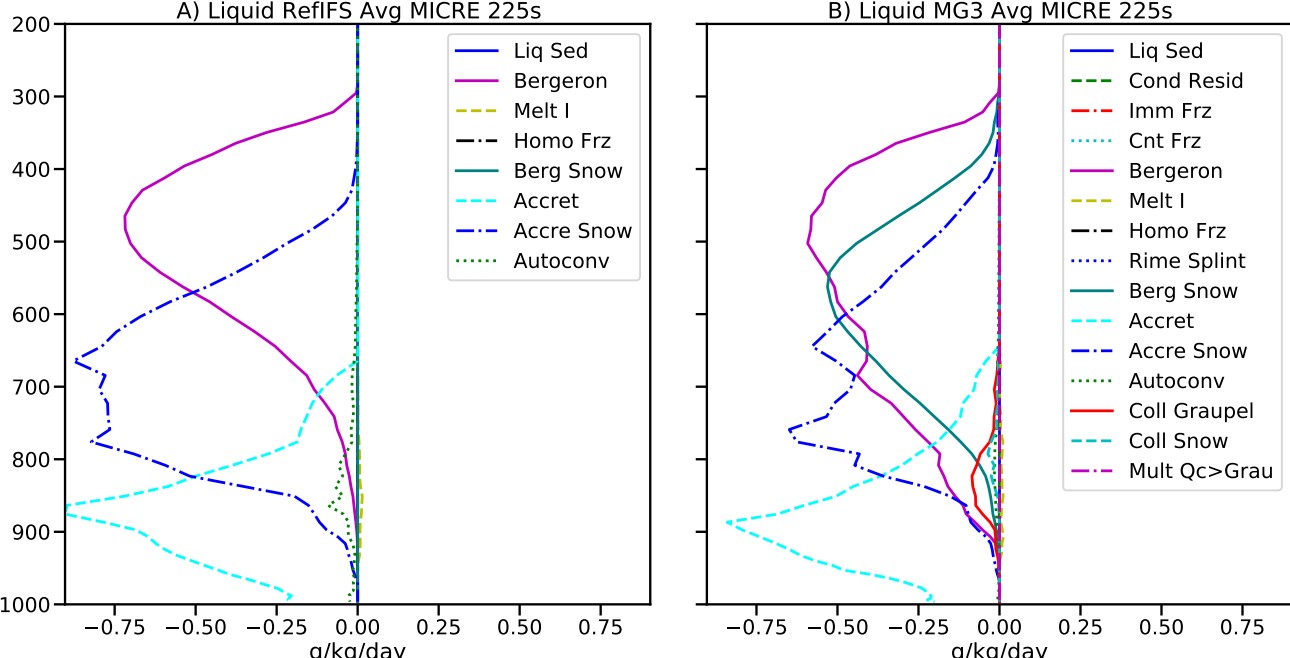

**Figure 7.** Profiles of cloud liquid water process rates, A) for the IFS microphysics and B) for MG3.

that at warmer temperatures. The major difference for liquid water is the vapor deposition process onto snow ('Berg Snow') in MG3; a process that is not active in the IFS. The IFS accretion onto snow compensates for the missing process to some extent, but still ends up less than the sum of the accretion and the Bergeron vapor deposition onto snow in MG3 at altitudes above 750hPa. There are several other terms that are different with relatively small magnitudes, such as the autoconversion of liquid

to rain, rime splintering and graupel collection of liquid.

    Ice process rates have similar dominant processes in both microphysics schemes (Figure 8), with a source from vapor deposition ('Bergeron') offset by a loss of ice through autoconversion to snow ('Autoconv Qi>Qs'). However, the Bergeron process is much more active at lower altitudes/warmer temperatures for MG3 compared to the IFS (as also seen in Figure 7) , with a similar difference in magnitude for ice autoconversion to snow which compensates. The more active Bergeron process

in MG3 is likely due to the larger supercooled liquid water present at these lower levels (Figure 6A). In MG3 (Figure 8B), there is also loss from accretion of ice onto snow ('Accret Qi>Qs') throughout the depth of the cloud from 300 to 800 hPa, and a source due to deposition onto ice ('Ice Sub/Dep') at high altitudes around 400 hPa. The ice sedimentation in the IFS shows a small loss in the upper part of the cloud and gain lower down, whereas the net redistribution is much smaller in MG3; a difference likely due to the different sedimentation numerics in the two schemes.

Rain process rates are illustrated in Figure 9. Rain sedimentation ('QR SED') is the major loss for precipitation, larger in the IFS, and snow melting ('Melt Snow') and accretion of liquid cloud droplets ('Accre') are the main rain source terms. Snow melting is more active nearer the surface in the MG3 scheme due to a deeper melting layer. There is not much rain evaporation





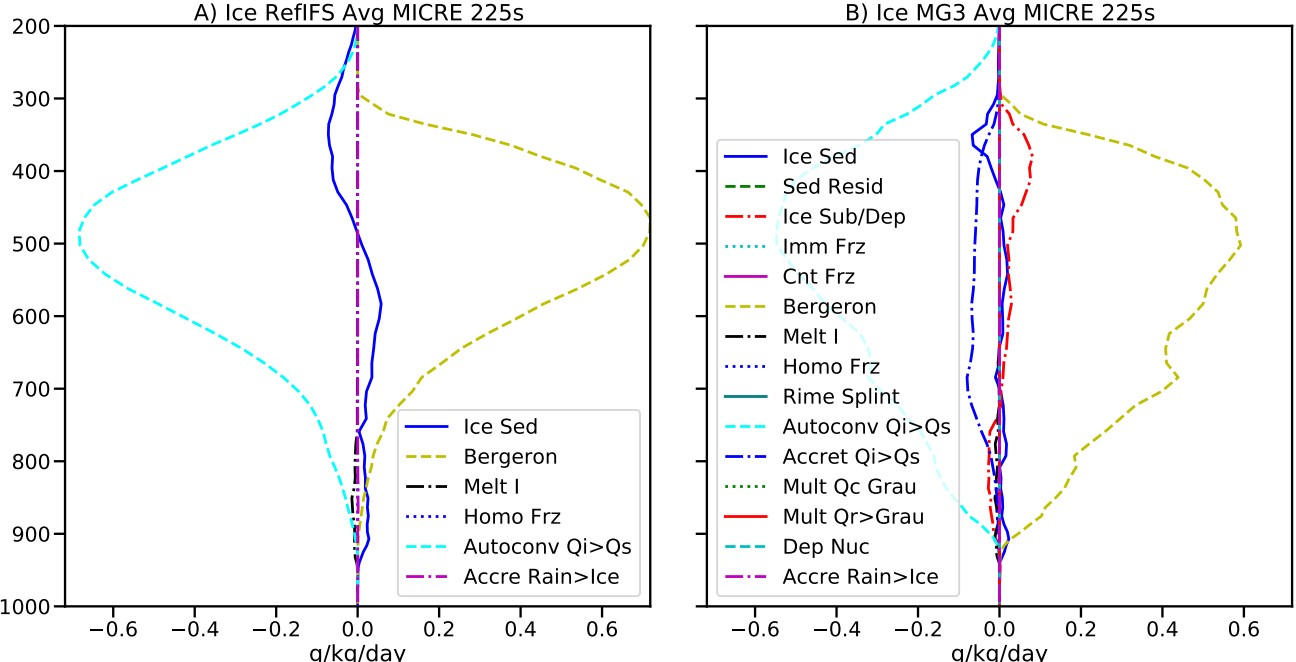

**Figure 8.** Profiles of ice process rates, A) for the IFS microphysics and B) for MG3.

in the PBL which is consistent with large reflectivity in the lowest kilometre (Fig 5). This is due to the specific rain evaporation assumptions in the IFS and modified MG3 (see section 2.3). There is even less rain evaporation in MG3 due to the deeper melting layer, and the two-moment scheme allowing rapidly falling rain drops rather than slower falling drizzle.

There are some differences in snow process rates (Figure 10). The general balance at upper levels is a sedimentation loss ('QS SED') and an ice autoconversion ('Autoconv Qi>Qs') source, representing aggregation of ice crystals. Snow fall speeds are fixed in the IFS and set to be the same in MG3, making the sedimentation rate similar. MG3 also has a source from the vapor deposition onto snow ('Berg > Snow'), and loss from snow sublimation/evaporation ('Evap') (larger in MG3 than IFS). At lower levels, the sedimentation source term for snow falling from above ('QS SED') is balanced by a melting loss term to rain ('Melt Qs'), which is just the inverse of the melting source term for rain (Figure 9). The shallower melting layer in IFS is also evident.

In summary, both microphysics schemes produce a largely similar balance of process rates. Given that some of the microphysical process formulations are similar between the two schemes and that there is some convergence that has been imposed by adjusting the MG3 scheme towards the IFS (see Section 2.3), this is perhaps not surprising. However, there are also some differences in the formulations that result in different hydrometeor profiles, particularly for the radiatively important super-cooled liquid water. As shown in Figure 6, the mean profiles of ice and snow are very similar between the two schemes (panels B,C). There is clearly compensation between some of the processes, such as the higher Bergeron process rate for ice growth and snow accretion rates in the IFS which compensates for the lack of a Bergeron process for snow in the IFS compared to





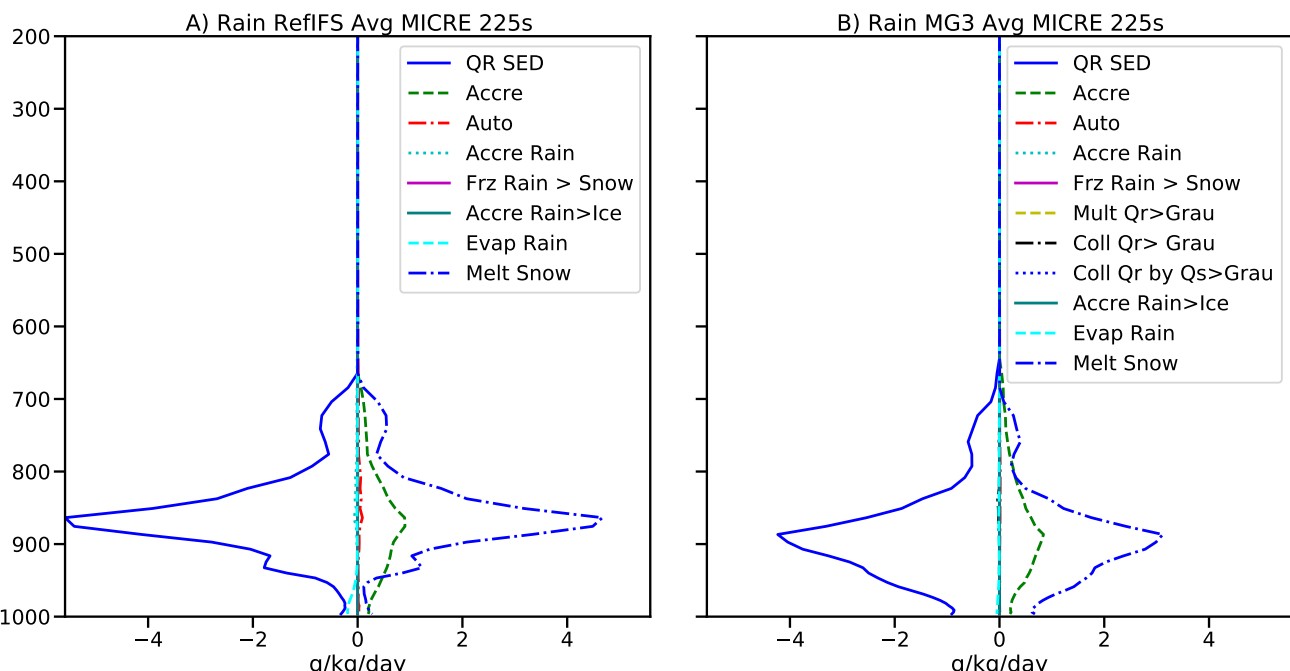

**Figure 9.** Profiles of rain process rates, A) for the IFS microphysics and B) for MG3.

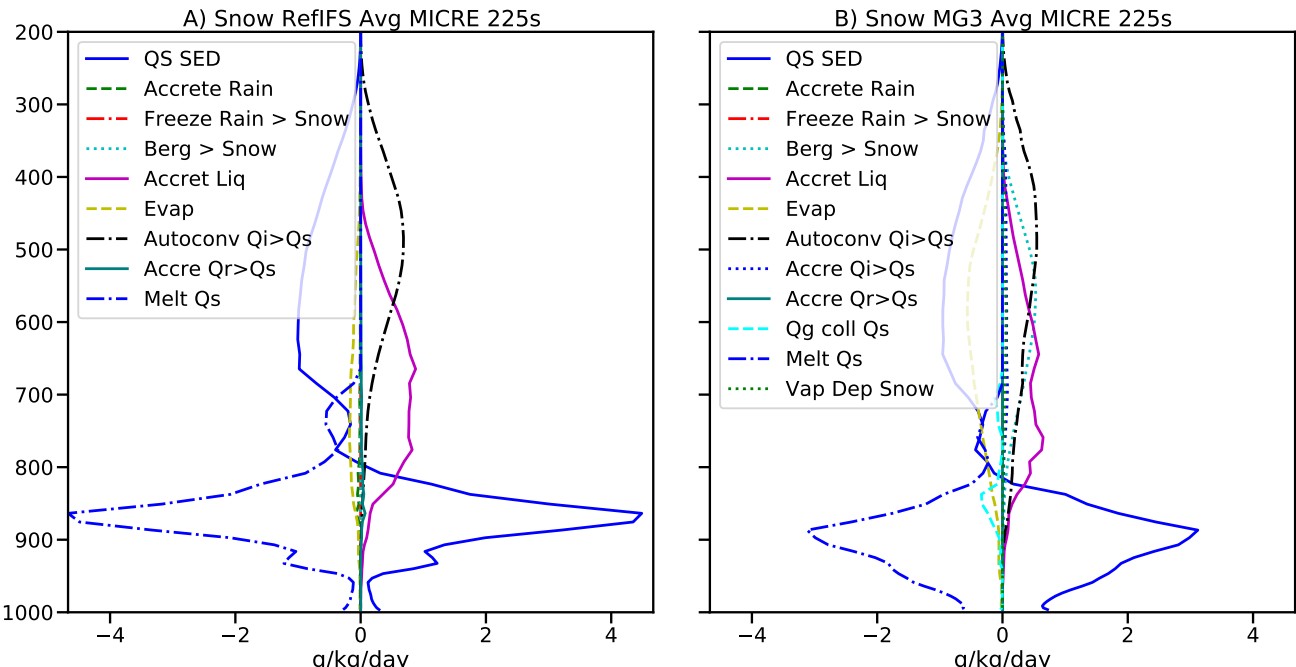

**Figure 10.** Profiles of snow process rates, A) for the IFS microphysics and B) for MG3.



MG3 (Figure 7). This compensation results in a similar supercooled liquid water profile at altitudes above 750hPa, apart from for the MG3 900s timestep simulation, (Figure 6A). However, at altitudes below 750hPa, the profile of supercooled liquid water is more than a factor of two different, with much higher values in MG3 compared to the IFS (Figure 6A). The higher cloud liquid accretion rates for both snow and rain below 750hPa in the IFS (Figure 7) are likely leading to the lower SLW in the IFS, which agrees more closely with the observations (Figure 2). At lower levels below 850hPa, there are also differences in rain in the precipitation profile (Figure 6C). For sedimenting precipitation, differences can come from higher in the column, but given that the ice and snow hydrometeor profiles are very similar, differences in the low level precipitation must therefore be due to the local impact of processes, particularly the melting rate of snow to rain, evaporation of rain, and differences in the rain fall speed.

### 3.2.3 Modifications and Sensitivity

In order to further understand the impacts of the changes to the MG3 scheme and differences with the IFS, a series of sensitivity experiments are performed changing individual aspects of the MG3 microphysics. The first set of changes (Figure 11) investigates the impact of the initial modifications from the original CAM6 version to the base MG3 scheme used for all the simulations described so far, as described in Section 2.3. These changes were made for this study to remove some of the most obvious differences due to numerical implementation and basic threshold and other assumptions, to allow a more meaningful comparison of the process rates between the two microphysics schemes. The second set of changes investigates the sensitivity to assumptions that affect several of the important microphysical process rates (Figure 12).

The sensitivity experiments are summarized in Table 1 and are described in more detail below. Figures 11 & 12 show the impacts on time-averaged vertical profiles for various quantities and the impacts on time-averaged single level quantities are shown in Figure 13. For reference, the figures also show the 'IFS', the modified MG3 used for this study ('MG3 Base'), on which all of the sensitivity experiments are based, and the original 'MG3 CAM6' settings used with the MG3 microphysics in the CAM6 atmospheric component of the coupled climate model CESM2. In general, the MG3 CAM6 settings are outliers for various quantities and far away from the IFS and 'MG3 Base' results. It is therefore notable that significant differences can occur from the numerical implementation of the sedimentation and microphysical threshold assumptions.

*(i) Sensitivity to MG3 modifications from CAM6*

We first assess changes to the fall speed of hydrometeors ('Fallspeed') and numerical formulation of the sedimentation of precipitation ('Explicit Fall') in MG3. The 'Fallspeed' sensitivity test removes a numerical fix for zero fall speed in the column when no precipitation is initialized there in MG3, and 'Explicit Fall' reverts the change from an implicit sedimentation of precipitation calculation, which is more stable across time-steps, back to the explicit scheme in CAM6. In addition, the IFS and MG3 Base have constant fall speeds for ice and snow, which are reverted to the MG3 CAM6 variable fall speed formulation in the 'Sediment' test. The 'Fallspeed' and 'Explicit Fall' slightly decrease the total amount of ice (Figure 11B) and the 'Sediment' change increases the (supercooled) cloud liquid water at altitudes above 700 hPa (Figure 11A). A reversion to allow the immediate evaporation of sedimenting condensate ('Evapsed') has negligible impact.



A series of changes to harmonize evaporation was also conducted ('Old Evap') as the IFS uses a relative humidity threshold of 90% before evaporation starts, and the IFS has a similar evaporation formula as MG3 but is multiplied by a factor of 0.3. The rain freezing threshold ('Rain Freeze') has small effects on total precipitation (Figure 11C) near 850hPa in the temperature range just below -5°C. Finally the IFS and MG3 Base do not have a threshold RH for ice nucleation (RH for IN) whereas MG3 CAM6 has an RH threshold of 105% with respect to ice for ice nuclearion to occur. Reverting the RH threshold for ice nucleation (RH for IN) increases liquid mass (Figure 11A) and number (Figure 12D) near 600hPa and reduces ice number (Figure 11E) significantly between 300–600hPa by making it harder to form ice. Each of these experiments is a single reversion experiment that removes these elements from MG3 base in IFS to get back to MG3 CAM6. However, the combination of these effects is important. The test 'Old Rain' combines the Rain Freeze, Sediment and Fallspeed tests, and it is this combination that reduces the unrealistic increase in precipitation (rain) mass near the surface in MG3 CAM6 (Figure 11C).

The implicit formulation for sedimentation as in MG3 Base has lower LWP (Figure 13A) than MG3 CAM6, and shortwave (Figure 13E) and longwave (Figure 13F) downward radiation more similar to the IFS which also has an implicit formulation. The largest difference between MG3 CAM6 and the MG3 Base is due generally to the different fall speed estimates (mostly affecting total ice) and rain freezing (affecting total ice and precipitation). Note that the SW downward radiation (Figure 13E) is generally inversely proportional to the LWP (Figure 13A) and to a lesser extent total cloud cover (Figure 13C). The spread in the LW downward flux is relatively small (6 W/m$^{-2}$, Figure 13F), relative to the spread in SW downward flux (35 W/m$^{-2}$, Figure 13E).

Finally, it is worth noting that supercooled liquid water above 700hPa is much higher in the MG3 CAM6 version (Figure 11A). This is due to the fall speed changes ('Sediment', which is also in the 'Old Rain' test) combining in a non-linear way with the change to the relative humidity used for ice nucleation ('RH for IN').

*(ii) Additional MG3 process sensitivity tests*

The second set of tests is performed with further modifications to the 'MG3 Base' code to understand its sensitivity relative to the IFS.

One of the most important aspects of the microphysics is the generation of precipitation. For liquid clouds, this is defined by the collision-coalescence process of interacting hydrometeors of different sizes. In bulk microphysics schemes like the IFS and MG3, this process has to be highly parameterized since the different sized drops are not explicitly represented. The parameterizations are for autoconversion (formation of rain from cloud water) and accretion (the removal of cloud water by rain). In MG3 and IFS, these parameterizations come from Khairoutdinov and Kogan (2000), a regression fit to a bin microphysical model. In MG3, the parameters for autoconversion were adjusted to better match recent observations, with resulting reduced sensitivity to droplet number, whereas in the IFS the original Khairoutdinov and Kogan (2000) parameters are used ('IFS KK'). In the IFS there is also an enhancement factor of 1.5 for autoconversion and 3 for accretion ('Ac3Au1.5') representing the effects of subgrid heterogeneity of cloud and precipitation. We test the MG3 scheme with these factors. The scaling of autoconversion and accretion ('Ac3Au1.5') significantly reduces cloud water, as expected, (Figure 13A) and slightly increases ice water path (Figure 13B), with corresponding increases in the downward SW radiation (Figure 13E). Changing



**Table 1.** Sensitivity tests applied to MG3 simulations

| Name | Description |
| --- | --- |
| MG3 Base | Base MG3 code |
| MG3 CAM6 | MG3 code with CAM6 settings |
| | Changes to revert MG3 Base to MG3 CAM6 |
| Fallspeed | Remove fallspeed correction that ensures non-zero sedimentation |
| Explicit Fall | Revert from implicit to explicit fallspeed |
| Sediment | Variable snow and ice fall speed |
| Evapsed | Allow evaporation of sedimenting condensate. |
| Old Evap | Original RH threshold for evaporation and un-scaled evaporation |
| Rain Freeze | Revert rain freezing temp to -40°C |
| RH for IN | Use 105% RH over ice threshold for ice nucleation (rather than no threshold) |
| Old Rain | Combines Rain Freeze, Fallspeed and Sediment |
| | Other Sensitivity Tests |
| Ac3Au1.5 | Sub-grid heterogeneity scaling for Accretion (x3) and Autoconversion (x1.5) following IFS |
| IFS KK | Use original constants for Autoconversion following IFS |
| Au in Ac | Allow accretion to see recently autoconverted rain |
| No Graupel | Graupel off (riming forms snow not graupel) |
| No Meyers | Removal of Meyers et al. (1992) ice nucleation |
| IN/5 | Ice nucleation of 1 $L^{-1}$ per timestep |
| IN*5 | Ice nucleation of 25 $L^{-1}$ per timestep |
| CCN*4 | Liquid activation rate of $4 \times 10^4$ $m^{-3}$ $s^{-1}$ |
| CCN/4 | Liquid activation rate of $0.25 \times 10^4$ $m^{-3}$ $s^{-1}$ |
| N=cnst | Constant number concentrations: $N_c$ = 200 $cm^{-3}$, $N_i$=50 $L^{-1}$, $N_r$=5 $L^{-1}$, $N_s$=1 $L^{-1}$, $N_g$=0.5 $L^{-1}$ |

MG3 to the IFS parameters for Khairoutdinov and Kogan (2000) ('IFS KK') has more of an impact higher up, reducing SLW
at altitudes above 800hPa, but with the major impact seen as a large increase in ice number (Figure 12E). This seems to result
from a conversion process from liquid (possibly freezing rain) to ice at supercooled temperatures. These settings produce a
simulation with LWP and surface SW downward radiation closer to observations (and IFS microphysics), largely due to lower
LWP (Figure 13A). We also test the MG3 scheme with a change that allows the autoconverted rain to be immediately used
by the accretion process ('Au in Ac'), which slightly enhances accretion but has little overall effect on the simulations. The
CAM6 settings result in the highest LWP (Figure 13A), which does enhance the surface LW down (Figure 13F) to be closer to
the observations (but note the difference is only a few $Wm^{-2}$), but produces much too small surface SW down (Figure 13E).

Two changes to the mixed phase have substantial impacts. Turning off the prognostic graupel (rimed ice particles) in the
MG3 simulation results in a significant increase in ice mass (Figure 12B) and precipitation (Figure 12C). This is due to the
removal of a loss process for rain onto graupel, which now occurs only onto snow, a less efficient process, and leads to higher





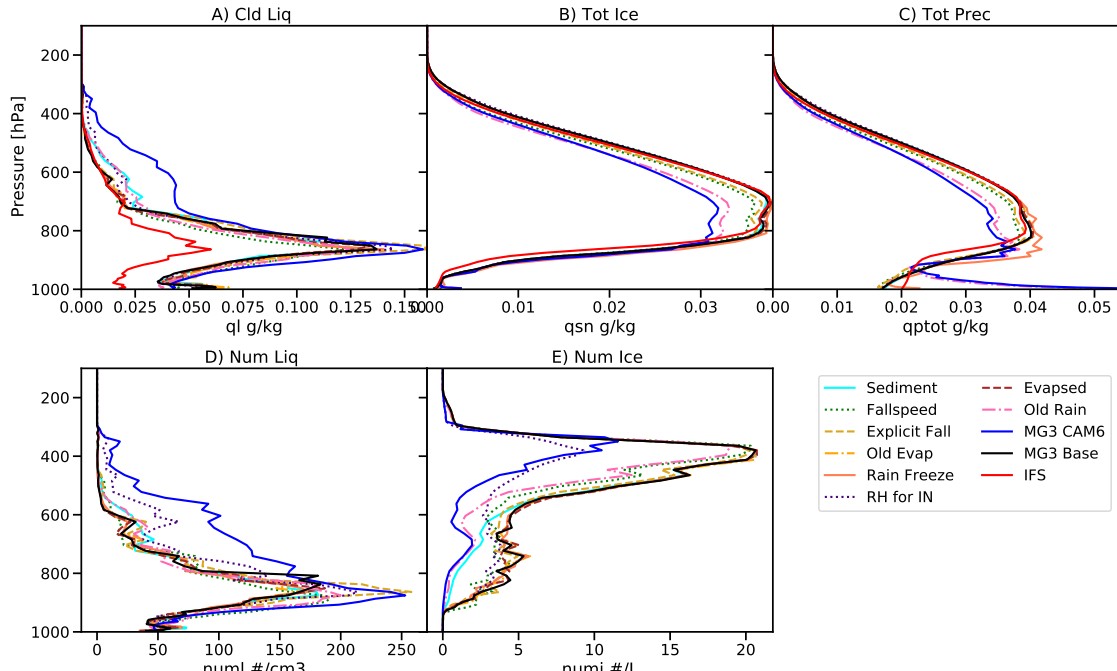

**Figure 11.** Vertical time averages of A) Supercooled liquid water, B) Total ice (ice+snow+graupel), C) Total precipitation, D) Liquid number concentration, E) Ice number concentration, for the MG3 base version (black solid) and various MG3 microphysics sensitivity tests as noted in the legend to differentiate MG3 and IFS microphysics. The reference IFS microphysics is included in A,B,C (red solid).

snow mass. Removing the Meyers et al. (1992) mixed phase ice nucleation significantly reduces ice number (Figure 12E) and IWP (Figure 13B), as well as supercooled liquid at higher altitudes (Figure 12A).

Next, we examine the role of number concentrations for liquid and ice in the simulations. Because MG3 typically runs with an aerosol model, it requires the activation of liquid drops and ice crystals to set the hydrometeor number concentrations, unless they are set to a constant. In the IFS there is no explicit representation of activation of aerosol, so for the MG3 simulations a

constant rate of cloud condensation nuclei (CCN) and ice nuclei (IN) production is defined. The ice and liquid mass and number concentrations are sensitive to these rates and this will naturally create differences between MG3 and IFS microphysics. Indeed, increasing the CCN production rate ('CCN*4') results in many more (Figure 12D) and hence smaller cloud water drops, increasing SLW (Figure 12A) and averaged LWP (Figure 13A), and reducing downward SW (Figure 13E). The opposite effects are seen for reductions in the CCN production rate ('CCN/4'). There are also tests performed adjusting the constant for

the rate of production of ice nuclei (IN/5 and IN*5). More IN slightly increases IWP (Figure 13B,C) with opposite effects for reducing IN.



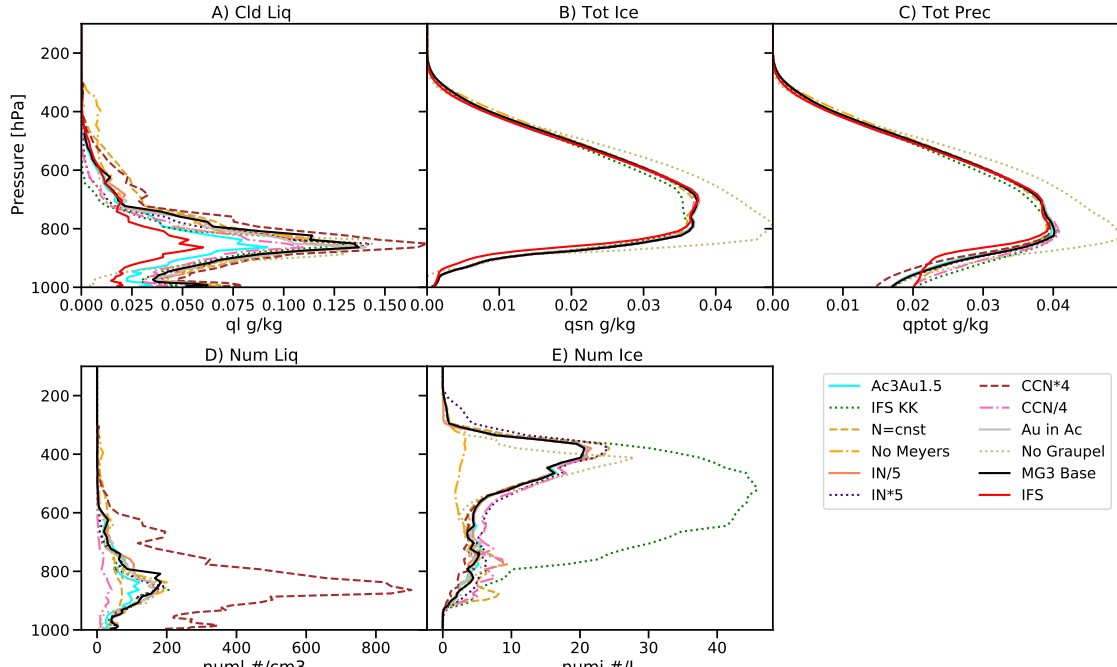

**Figure 12.** Vertical time averages of A) Supercooled liquid water, B) Total ice (ice+snow+graupel), C) Total precipitation, D) Liquid number concentration, E) Ice number concentration, for the MG3 base version (black solid) and various MG3 microphysics sensitivity tests as noted in the legend to understand MG3 sensitivity.

We also perform a simulation with fixed number concentrations for all species in the MG3 scheme (rather than fixed production rates), as specified in Table 1. Fixed number concentration ignores any drop or ice nucleation, resets the prognostic number to a set value every timestep, and uses this throughout the microphysics. Fixed number concentrations ('N=cnst') have a sub-
stantial effect on the simulations if they are not set to reasonable values. Here the number concentrations have been adjusted to produce similar number concentrations as the prognostic MG3 Base simulation. Nonetheless, even with similar LWP, average liquid number concentrations are decreased (Figure 12D) similar to the CCN/4 experiment. Note that the average number is lower than the fixed drop number due to averaging and number depletion.

## 4  Summary and Conclusions

Simulations are performed with the IFS single column model with two "operational" microphysics parametrizations (the one-moment IFS scheme used at ECMWF for NWP, and the two-moment MG3 scheme used in the CAM6 climate model). Both are



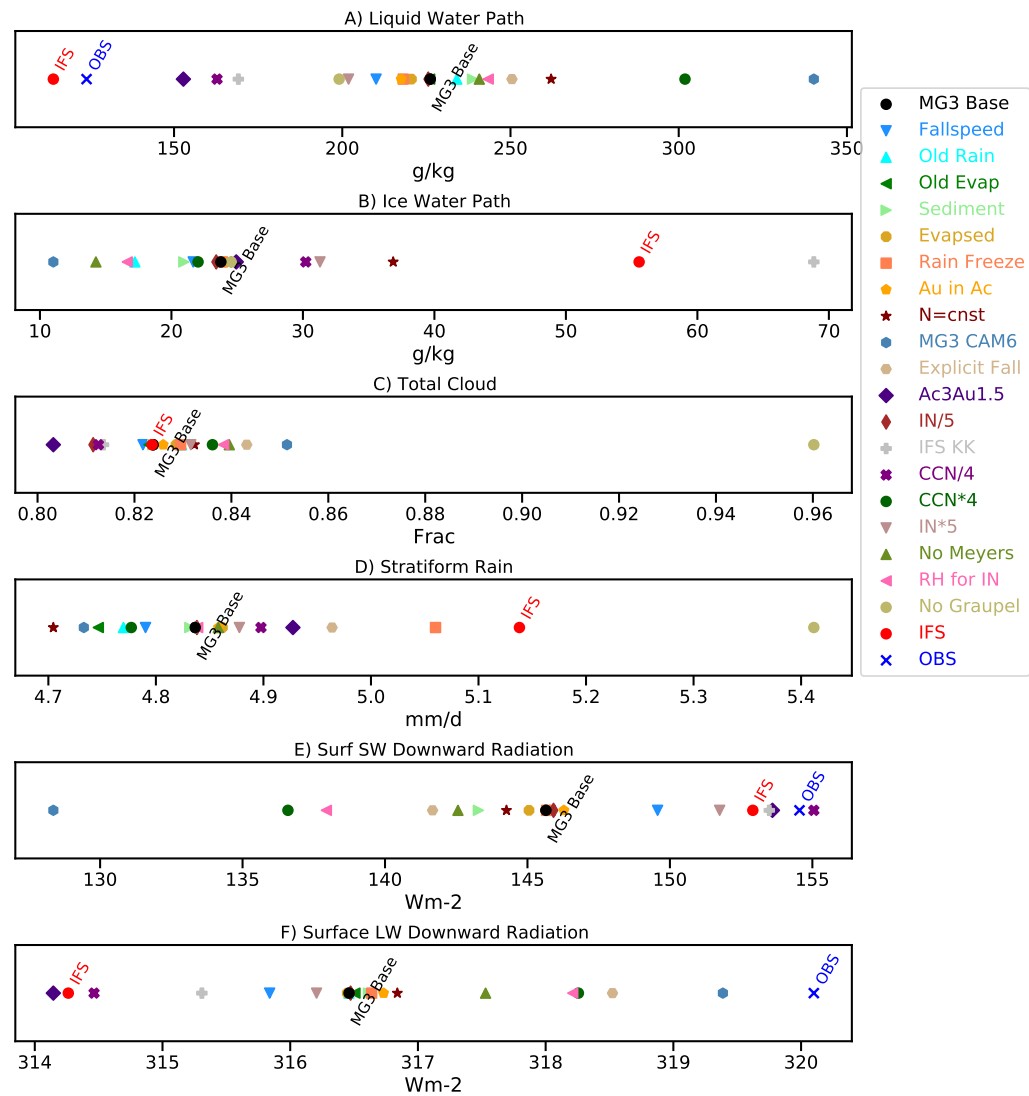

**Figure 13.** Time averages of A) Liquid water path, B) Ice water path, C) Cloud fraction, D) Surface stratiform rain rate, E) Surface SW down and F) Surface LW down for MG3 microphysics sensitivity tests as noted on the plot. MG3 base case in black. Reference IFS plotted in Red. MICRE observations for LWP, and SW and LW radiation in Blue.



able to reproduce many of the characteristics, evolution and time-mean variability of Southern Ocean clouds observed during the MICRE campaign at Macquarie Island during the period January–March 2017, in a regime with significant amounts of supercooled liquid cloud.

Surface radiation is well represented compared to the ground-based radiometer and satellite data sets with radiative flux biases within or close to the uncertainty of the observations in both the LW and SW. In particular, in the shortwave solar spectrum, there is no "too few/too bright" problem that has previously been seen in some global models (Wall et al., 2017; Kuma et al., 2020). Hydrometeor occurrence (cloud fraction) is slightly higher than observed when radar simulator output is compared to observations.

There are large uncertainties in the observed surface (passive microwave) LWP retrievals, largely due to the presence of precipitation, but for low level non-precipitating clouds, the surface radiometer values at Macquarie Island are consistent with satellite estimates based on geostationary (Himawari-8) and MODIS polar orbiting satellites. Overall, the mean LWP for the IFS microphysics compares well to both surface and satellite LWP retrievals for low clouds, while the mean LWP from the MG3 microphysics is too large. An analysis of the distribution of LWP shows that the MG3 microphysics produces more high

LWP events than the observations and IFS simulations, but the median LWP values for MG3 are comparable to the IFS and the observations. Because the radiative effects of condensed water saturate when the LWP is larger than $\sim 400 \text{ g m}^{-2}$ such that cloud albedo does not change much, the large LWP events do not translate into a large SW radiation bias in the MG3 scheme.

 A comparison of radar reflectivity with the vertically pointing radar at Macquarie Island shows the simulations capture the overall characteristic shape of the reflectivity-height distribution, although the mean frequency of occurrence of hydrometeors

(above the surface radar minimum detectable signal) is too high, at least above the boundary layer. In general, the simulations contain a higher occurrence of reflectivity factors larger than -15 dBZ compared to the observations. While some of this difference is expected due to attenuation of the observed radar reflectivity that results from rain on the radar radome, the difference also occurs for periods when there is no surface rain. Rather this difference in the distribution of reflectivity factors is largely driven by the model assumption that when precipitation is present, it occupies the same area (fraction) as that

predicted for cloud, and a better sub-grid representation is needed in the model microphysical schemes in order to addresses this "too frequent, too light" precipitation problem. While this problem is certainly not unique to the Southern Ocean (Stephens et al., 2010), it is perhaps especially pertinent because of the high occurrence of light precipitation in the region.

 The single column model is a partially constrained testing environment and for long simulations requires forcing to keep the temperature, humidity and wind profile from drifting too far from the observations. However, the cloud and microphysical

response to the forcing is not constrained and thus it is a useful platform for assessing different microphysical schemes. In a test without any relaxation of the temperature, the PBL deviates from observations, and significant SW and LW radiation biases emerge on the order of tens of $\text{W m}^{-2}$ with both the IFS or MG3 microphysics. Clearly, part of getting clouds and cloud radiative effects correct is getting the PBL structure right and the correct coupling of the thermodynamics to the cloud physics, as well as the microphysical processes themselves.

A series of sensitivity tests modifying the numerical and physical representation of various cloud and precipitation processes in the MG3 scheme are performed to understand the differences from the IFS microphysics scheme. The 'MG3 Base' scheme





used for the comparison with observations contains an initial set of changes to bring MG3 closer to the IFS, so that the focus of the study can be on differences due to the individual microphysical processes, rather than the differences in numerical implementation of sedimentation, certain hardwired threshold values, or the lack of prognostic IN/CCN in the IFS. With this modified 'MG3 Base' microphysics scheme, the average ice, snow and rain hydrometeor profiles are remarkably similar, with the main difference being double the amount of supercooled liquid water on average in MG3, as seen in the comparison against observations. The sensitivity simulations show that differences in the autoconversion to rain and accretion onto rain loss processes for liquid water at sub-freezing temperatures are the primary cause of these differences, and the MG3 scheme could be adjusted to enhance the liquid loss process and reduce the LWP in deeper clouds.

The required complexity of microphysical parametrization is a key question for future development of atmospheric models, and one of the motivations for the comparison here of the one-moment (mass) IFS with the higher complexity two-moment (mass and number concentration) MG3 scheme including graupel. Representing a graupel (rimed ice) hydrometeor in MG3 has a small impact with more precipitation and less SLW than without graupel, but there was not a significant amount of graupel present in the type of clouds at this location and a larger difference would be expected in regimes with more deep convection. Regarding the two-moment versus one-moment representation, it is difficult to disentangle the impacts on the mean fields of the two-moment representation in MG3 from the differences in microphysical process formulation between the schemes as there can be compensation between processes, and numerical formulation and different tunings of microphysical process rates can dominate. However, despite similar ice and precipitation mass profiles, there are differences in the radar reflectivity profiles which are at least partly due to the additional degrees of freedom from the representation of particle number concentration in MG3. Finally, for cloud liquid and cloud ice formation, there are clearly large sensitivities to the specification of the CCN and IN production rates, and although tuning these in the SCM can bring closer agreement, a full comparison in the 3D model with interactive aerosol activation is required for the impact to be fully assessed.

In summary, this study provides an evaluation of two operational microphysical parametrization schemes, one from the CAM6 climate model and one from the ECMWF global NWP model, within the IFS single column model framework for Southern Ocean clouds with significant supercooled liquid. Further work is underway to introduce graupel and particle number concentrations into the IFS microphysics towards a similar level of complexity as the MG3 scheme. Given the sensitivities to the number concentration and activation (especially of ice) it is critical to properly represent these processes for an accurate simulation of Southern Ocean ice and liquid clouds and their important role in the global radiation budget. The use of a wider range of in situ and cloud-sensitive satellite observations will be crucial to constrain both mass and number concentrations. A future direction will be full three-dimensional tests within the IFS assimilation and forecast system to improve our understanding of the role of additional complexity in microphysical parametrization on the realism and skill of global weather forecasts.

*Code and data availability.* Simulation output used in this manuscript, and specific code for MG3 microphysics in the IFS is available in a zenodo archive. DOI:10.5281/zenodo.10037746 (https://zenodo.org/records/10037746). Data used for comparison is available through the



noted references in the text. All MICRE data are available from the Atmospheric Radiation Measurement (ARM) data archive (www.arm.gov), a U.S. Department of Energy (DOE) Office of Science user facility managed by the Biological and Environmental Research Program. Specifically, data from MICRE can be found using the data discovery tool for the "MCQ" site.

    The IFS source code is available subject to a licence agreement with ECMWF. ECMWF member-state weather services and approved partners will be granted access. The OpenIFS single column model code is also available for educational and academic purposes via an

OpenIFS licence (see http://www.ecmwf.int/en/research/projects/openifs).

*Author contributions.* AG, CCC, RF and MF developed and evolved the MG3 microphysics code for the open IFS. AG performed the simulations. Analysis and figure development were conducted by AG, RM, RF and MF. AG wrote the manuscript with editorial input and sections written by all.

*Competing interests.* The authors state there are no competing interests

*Acknowledgements.* The Pacific Northwest National Laboratory is operated for the U.S. Department of Energy by the Battelle Memorial Institute under contract DE-AC05-76RL01830. This work was supported in part by the U.S. Department of Energy Atmospheric System Research program through Grant DE-SC0016225. The National Center for Atmospheric Research is supported by the U.S. National Science Foundation.



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
