# Peer review of "The Impact of Cloud Microphysics and Ice Nucleation on Southern Ocean Clouds Assessed with Single Column Modeling and Instrument Simulators"

_EGUsphere, 2024_

## Referee Comment (RC1)

Review of "The Impact of Cloud Microphysics and Ice Nucleation on Southern Ocean Clouds Assessed with Single Column Modeling and Instrument Simulators"

By Andrew Gettelman, Richard Forbes, Roger Marchand, Chih-Chieh Chen, and Mark Fielding

MS No.: egusphere-2024-599

**Recommendations: Major revision**

**General comments :**

This study compares the microphysical representations of the two schemes implemented to the ECMWF's global model, Integrated Forecast System (IFS), and those of the observation obtained by the Macquarie Island Cloud and Radiation Experiment (MICRE). A single-column model experiment is conducted using the tendency obtained by the global model. The two schemes compared are the scheme used in IFS (IFS) and the two-momentum scheme developed and improved by the authors (MG3). Finally, the authors compared the different implementations of MG3.

The evaluation and the improvement of the cloud microphysics scheme by comparing with the observation and the other schemes are continuously demanded to achive better representation of cloud distributions and characteristics in atmospheric models. A variety of observational data can be chosen for the model evaluation. The authors clarified the uniqueness and importance of the observation used in this study for the evaluation of supercooled liquid clouds, which are common at higher latitudes.

The validity of the use of a single column model (SCM) for testing cloud microphysical schemes must be more explained. SCM is widely used but has limitations. It is generally known to be useful for testing convection whose main process is vertical mixing. If it is used for a mid-latitude weather system, horizontal advection is a main source of the tendency. I do not understand whether in this approach, the hysteresis of the cloud microphysics processes is used for the update of SCM.

This study only compared a single location. I like this methodology, but it is not well understood whether the validation at a single location is generalized to other areas. The authors should discuss how the generalization and the universality of the cloud microphysics evaluation.

The upward-pointing cloud radar is severely attenuated under precipitating or convective clouds. Radar signal comparisons shown in Figures 4 and 5 should be carefully re-examined. The authors discuss upper cloud properties, but the radar signal above 5 km altitude is not reliable. On the other hand, the numerical model results with a simulator show distinct distributions in the upper layers. I suspect that the attenuation of the radar simulator is not properly calculated.

In the sensitivity experiments in Section 3.2.3, the authors pointed out the severe dependency on the assumption of cloud fraction and sub-grid decomposition, as well as the timestep. It is generally recognized that it is not straightforward to evaluate the cloud properties of GCMs. The authors should summarize which properties can be evaluated, and the others are more dependent on the model assumptions. This type of caution should be emphasized in the introduction and the discussion sections. The authors may also note differences in the methodology from the cloud evaluation of global storm-resolving models.

**Specific comments :**

p. 3, L 58: The type of the radar (W-band) and the frequency must be indicated in this section.

p. 4, Section 2.2.2 and 2.2.3: Specify the method or the definition of the cloud fraction. What are the assumptions of the size distribution? I suggest the full details of the assumptions of the size distributions of all the hydrometeors should be tabulated. The results of the simulator severely depend on them.

p. 5, Section 2.2.4: What is the methodology of the treatment of the sub-columns? Some are described in a later section. This section is the right place to be fully explained.

p. 5, L 128-131: As pointed out in General comments, the result from the attenuation calculation of the radar simulator is not understandable. The authors should quantitatively show the quality of the radar simulator in terms of the attenuation.

p. 6, L153, "the formation of layers with zero fall velocity": I do not understand what "zero fall velocity" means. Are they cloud water or cloud ice, which possesses zero fall velocity?

p. 6, Figure 1: It isn't easy to distinguish the differences among the results from this figure. The authors could consider a clearer method to show the difference. A possible improvement is to show the difference from the observation for the two numerical results in separate figures.

p. 11, L251, "The median values of MG3 LWP are comparable to those for the IFS and the observational datasets.": The median value of MG3 LWP is 220, and that for the IFS is 69.8. They are not comparable.

p. 11, L267, "a greater occurrence of low reflectivity in the upper troposphere related to ice cloud": The radar signal is highly attenuated for the upper troposphere. How reliable are the upper tropospheric clouds?

p. 11, L268, "reflectivity-height histogram": It is generally called a Contoured Frequency Altitude Diagram (CFAD).

p. 11, L272 and after, "left" and "right": These should be smaller and larger values, respectively.

p. 11, L274-275, "The model simulations contain a large occurrence of hydrometeors above 5 km that the surface radar would be unable to detect.": Why so many smaller values of radar reflectivity are detected above 5 km for MG3 and IFS (Fig. 5)? This result suggests that the simulator's calculation of attenuation is not adequate.

p. 11, L280, "This suggests that the MG3 is likely better representing this high-altitude cloud.": The high clouds are severely affected by the model tuning. In which aspect the high cloud is better represented?

p. 14, L285, "affected by the model sub-grid representation": It must also be affected by attenuation, I guess.

p. 14, L285-307: The description of this part illustrates the difficulty in assessing the nature of GCM clouds. When evaluating them by comparison with observations, the cloud properties in the GCMs are greatly influenced by the assumptions used in the GCMs, and thus, the assumptions themselves are evaluated in some cases. This problem illustrates the difficulty of this study and could be emphasized more. The definition of cloud cover and the subgrid model assumptions are important elements of this uncertainty. In the introduction and the discussion section, this issue, in particular, must be highlighted.

p. 14, L314-315, "the analysis shows the total low cloud cover in the simulations is reasonable": Please show this result.

p. 15, L336-337, "There is overall more supercooled liquid water": Please indicate where is the location of the 0-degree altitude.

p. 15, L339-340, "A common problem is a sensitivity of microphysical processes to the length of the timestep": This serious problem is pointed out abruptly. Do the authors summarize the artificial factors that severely affect cloud characteristics? I speculate that the IFS is not so much affected by the timestep.

p. 15, L349, "supercooled liquid water (SLW)": SLW is already defined at L337. It does not need to be spelled out here.

p. 17, L353, "by regime": What is the regime?

p. 18, L377, "a deeper melting layer": Where is the melting layer? How deep is the melting layer of the MG3 scheme?

p. 21, L425, "the MG3 CAM6 variable fall speed formulation": What is the detail of the formulation of the fall speed?

p. 25, before the end of Section 3.2.3: A lot of the sensitivity results are shown. The authors should summarize the sensitivity experiments by itemizing the main results or in a table quantitatively.

p. 27, L515, "above": Is this higher in altitude or larger in signal?

p. 27, L517, "While some of this difference is expected due to attenuation of the observed radar reflectivity that results from rain on the radar radome": This caution must be explicitly described in the main text.

p. 27, L523-329: This paragraph explains the usefulness of the single-column model. I agree. However, I suspect that there is a limitation or caution if it is applied to mid and high-latitude areas.

p. 28, L537-539, "The sensitivity simulations show that differences in the autoconversion to rain and accretion onto rain loss processes for liquid water at sub-freezing temperatures are the primary cause of these differences, and the MG3 scheme could be adjusted to enhance the liquid loss process and reduce the LWP in deeper clouds.": If the authors claim that this part is the main conclusion, it should be more emphasized when summarizing the sensitivity results in Section 3.

---

## Referee Comment (RC2)

**Review of "The Impact of Cloud Microphysics and Ice Nucleation on Southern Ocean Clouds Assessed with Single Column Modeling and Instrument Simulators"**

Using an ECMWF SCM, the paper compares two different cloud microphysical schemes, MG (double moment) and IFS (one moment), to ground and satellite based observations over a 2.5 month period (January 1 - March 17, 2017) during MICRE. Large differences of water mass and reflectivity between the schemes are found. Many sensitivity experiments on the cloud microphysics schemes found that Surface radiative fluxes and total water path are highly sensitive to the formation and fall speed of precipitation. Although MG produces well the observed reflectivity and radiative fluxes, higher liquid water contents is seriously overestimated. The sensitivities of the results to liquid Cloud Condensation Nuclei (CCN) and Ice Nuclei are also investigated.

The paper is well written and the results are interesting and important for model improvement.

**Recommendations**

Minor revisions

**Minor points**

1) The overestimation of the LWP by MG is too large. EXP Ac3Au1.5 produces less LWP, but still about two times of IFS. Since the single moment can be treated as a special case of double-moment, could an experiment be designed to produce the similar LWP of IFS microphysics?
2) EXP "No Graupel " produces large precipitation, total Ice, and even the liquid is not small, where are the sources of those hydrometers?

---

## Referee Comment (RC3)

**Review of "The Impact of Cloud Microphysics and Ice Nucleation on Southern Ocean Clouds Assessed with Single Column Modeling and Instrument Simulators"**

by Andrew Gettelman, Richard Forbes, Roger Marchand, Chih-Chieh Chen, and Mark Fielding

MS No.: egusphere-2024-599

**Recommendations: Major Revision**

**1   General Comments**

This paper describes an experiment comparing two microphysics schemes of differing complexity by testing in a constrained single-column model framework. The results were compared against field observations over an extended period of 75 days. The observations were made in the Southern Ocean where mixed phase cloud processes are important feature of the atmospheric system, and one which is currently poorly represented in numerical weather prediction models. As such this paper makes a useful contribution to the field by evaluating how both a single-moment (1M) and double-moment (2M) microphysics scheme performs in this region. The work takes a broad overview across the full time range, giving an insight into the problem, whilst also looking in detail at the diurnal cycle. Detailed study of various process rates in the 1M and 2M schemes are trialled and compared.

The observations data come from a 75 day period in Austral spring, on a remote Island in the Southern Ocean. Instrument simulators are employed to convert the model predictions into a form that can be directly compared to observations, e.g. hydrometeors are passed through a radar simulator code.

The major conclusions are that both the 1M and 2M schemes have some skill in representing the observed cloud and rainfall properties, both at the surface, top of atmosphere, in in the vertical cloud column, especially for bulk properties such as mass concentrations and liquid water paths (LWPs), including supercooled liquid water (SLW). The 2M scheme is shown to add value for radar reflectivities - hence clouds structures, but at the expense of somewhat overestimated LWP.

The results of a suite of tests are then presented, including time step sensitivity, process rate comparisons between schemes, and numerical formulations of some aspects of the microphysics - namely sedimentation, along with sensitivity tests of number concentrations of cloud condensation nuclei (CCN) and ice nucleating particles (INP).

The paper could be improved by adding clarity and perhaps a re-formulation of the aims and conclusions of the paper. It is not clear who the intended audience is, either Southern Ocean specialist, cloud model developers or modelling centres (numerical weather prediction (NWP) or climate) wanting guidance when choosing between 1M and 2M cloud microphysics. It might be all three but, some rewriting to tighten up the aims and conclusions is suggested. For example - Line 71 suggests

there will be guidance on where 2M schemes may be necessary - but this isn't directly concluded upon later. it might be the case (as is in fact stated by the authors) that full 3D simulations are required to go beyond the results in this work, something which would clearly require a step-change in compute resource. As well as highlighting that changing various parameters and assumption(e.g. number concentrations of aerosols) can alter the results, the work would benefit from a deeper commentary looking at the relative importance of the various aspects tested. otherwise this paper reads like a description of a test framework, as a forerunner to full 3D model simulations.

**2 Specific Comments**

**2.1 Title**

Mention of the comparison of 1M and 2M microphysics schemes would be more informative of the content of the paper. Also mentioning that the key difference is enhanced SLW in the MG3 2M scheme would point to one of the key results.

**2.2 Abstract**

it is stated that both dynamic and aerosol through microphysics are important, but the model tests only use single-column, and so it isn't clear what the "dynamic" refers to here - is it just the tests where the relaxation to prescribed temperature was turned off?

**2.3 results**

As a general comment - can the number of figures be reduced? For example, as mentioned elsewhere - can Fig. 6 be combined with 11 and 12 as anomalies?

Are all process rate figures needed (Figs. 8, 9, 10, 12)- they are also difficult to read. perhaps there could be a cloud liquid ice 2x2 and a recipitation rain and snow 2x2 panel?

– Lines 186-198 are repetition with the figure caption, read like methodology, and could possibly be tidied up or consolidated.

– Lines 212-216 - Stated that there is too much high cloud in IFS, resulting in poor TOA LW performance. this is followed up on in Sect. 3.1.3. it is mentioned that there are Line 277 states that IFS produces more of the low reflectivity values at altitude than MG3. But IFS also has a shoulder of higher reflectivity values (Fig. 5 middle right) compared to MG3 - could these values not be responsible for the poor TOA LW performance?

– Lines 229 onwards - There is discussion around the limitation of CERES SYN at night - should this data even be shown (possibly yes, to highlight the issues)?

– The use of MODIS at high latitude is interesting, and shows some evidence of a diurnal cycle - soon after sunrise - is this also an artefact, or believable? There is not much discussion on this particular aspect.

- Line 334 - A paragraph here to introduce the sensitivity tests in Sect. 3.2 would aid clarity of the following sections.

- Line 405 - the timesteps tests for IFS seem unnecessary.

- Figures 7, 8, 9, 10 - can resultant budgets be included here? For example Line 358 onwards, discussion of Bergeron process indicates that some processes not fully compensated for between IFS and MC3 - this could be shown visually too.

- Section 3.2.2 - authors comment that there aren't huge differences in outcomes, perhaps only that MG3 has more SLW. Does this section add a huge amount to the overall paper?

- There are significant differences in model results when scaling either CCN or INP by factors of 4, as might be expected, but it's not clear if these values a physical / climatologically reasonable or just for illustration. Do these test point to the need for use of 2M in NWP and climate simulations or not? Varying parameters individually will give different results, but the true test is covarying parameters, in a 3D model - which whilst commented on in the summary, does beg the question as to what these tests are trying to illustrate.

**2.4 Summary and Conclusions**

This section would benefit from editing, in particular to link more closely to the abstract and aims. One aspect seems to be the importance of sub-grid representation of precipitation and cloud fraction, which is talked about in the results text.

The paper shows that both IFS and MG3 can represent portions of the cloud fields, that surface radiation is OK, and that where data are good the LWP predictions are good, although perhaps MG3 is too large, even though this doesn't feedback on to SW biases at the surface.

**3 Technical Corrections**

- Line 205 - "TOA flux" should be "TOA SW Flux"

- Units should be represented as negative powers where appropriate and have a space between elements

- Line 360 - incorrect abbreviation of 'accretion' (also elsewhere, plus others not defined)

**3.1 Figure comments**

**3.1.1 General**

There are numerous formatting styles in use for figures in this manuscript and general inconsistencies that could be rectified.

- Many multiple panel plots need sub-panel labels, (a), (b), (c), etc, and the text updating accordingly

- Suggest that panel titles are removed and moved into the captions

- suggest moving variables names and units from panel titles onto y-axes

- units should be written as negative powers as appropriate, and spaces between elements (also in the body text)

- units should be enclosed in square brackets consistently

85
- Figures 6, 11, 12 only have y-axis titles on the top row

- Figure 2 needs the dashed-line describing in the caption

**3.2 specific**

- Figure 1 (and other time series) - could the time units be in days?

- Figure 1 - can the xtick but in 6 hourly (or 4?) intervals for 24 hours?

90
- Could Figure 6 be combined with Figs. 11 and 12 in to a 5x3 panel plot, and the figures 11 and 12 presented as anomalies, as the results are very difficult to visualise

- Figures 8 through 13 - many of the abbreviations of process rates are not defined, or incorrectly defined (typographical errors). Could a lookup table be provided perhaps?

---

## Author Comment (AC1)

**Replies to Reviewers**

**Reviewer #1**

General Reply: We thank the reviewer for their careful read of the manuscript. The suggestions and clarifications have greatly improved the manuscript. In response to these points, we have added further discussion and clarification of the comparisons between the radar simulator and the observed radar in several places. We have made all the other changes requested, except for presenting the sensitivity tests as difference plots. As noted, we made these figures and they were hard to interpret.

Review of "The Impact of Cloud Microphysics and Ice Nucleation on Southern Ocean Clouds Assessed with Single Column Modeling and Instrument Simulators"
By Andrew Gettelman, Richard Forbes, Roger Marchand, Chih-Chieh Chen, and Mark Fielding
MS No.: egusphere-2024-599
Recommendations: Major revision

General comments :
This study compares the microphysical representations of the two schemes implemented to the ECMWF's global model, Integrated Forecast System (IFS), and those of the observation obtained by the Macquarie Island Cloud and Radiation Experiment (MICRE). A single-column model experiment is conducted using the tendency obtained by the global model. The two schemes compared are the scheme used in IFS (IFS) and the two-momentum scheme developed and improved by the authors (MG3). Finally, the authors compared the different implementations of MG3.

The evaluation and the improvement of the cloud microphysics scheme by comparing with the observation and the other schemes are continuously demanded to achive better representation of cloud distributions and characteristics in atmospheric models. A variety of observational data can be chosen for the model evaluation. The authors clarified the uniqueness and importance of the observation used in this study for the evaluation of supercooled liquid clouds, which are common at higher latitudes.

The validity of the use of a single column model (SCM) for testing cloud microphysical schemes must be more explained. SCM is widely used but has limitations. It is generally known to be useful for testing convection whose main process is vertical mixing. If it is used for a mid-latitude weather system, horizontal advection is a main source of the tendency. I do not understand whether in this approach, the hysteresis of the cloud microphysics processes is used for the update of SCM.

>>The SCM trades a bit of fidelity in the advection of hydrometeors for the benefit of being able to control the thermodynamics of the simulation that govern cloud microphysics. Here, the representation of temperature and wind is more important than any small advection of

hydrometeors.  We have added a discussion of this point now to the conclusions to illustrate this.

This study only compared a single location. I like this methodology, but it is not well understood whether the validation at a single location is generalized to other areas. The authors should discuss how the generalization and the universality of the cloud microphysics evaluation.

>> As noted, one of the main points of this paper is to focus on S. Ocean clouds, which are known to create significant biases in weather and climate models. Added a bit of discussion to the methodology and the conclusions describing the representativeness of the location.

The upward-pointing cloud radar is severely attenuated under precipitating or convective clouds. Radar signal comparisons shown in Figures 4 and 5 should be carefully re-examined. The authors discuss upper cloud properties, but the radar signal above 5 km altitude is not reliable. On the other hand, the numerical model results with a simulator show distinct distributions in the upper layers. I suspect that the attenuation of the radar simulator is not properly calculated.

>> The comparisons between the radar simulator and the observed reflectivity are difficult, and for reasons beyond just attenuation of the single. Attenuation is due to the physical scattering of the radar signal by hydrometeors and gasses. This is what the radar simulator does for attenuation. The comparisons are also affected by several other factors beyond attenuation. One is that the radar has a minimum detectable signal with height. Another is that rain on the radome can also disrupt the signal. Finally, the cloud overlap assumption and any sub-grid assumptions about hydrometeors affect attenuation calculations. We have now clarified this when introducing the radar simulator, and in the text. There is a discussion of these issues as well in analysis of figure 4 and figure 5. Indeed, part of the point of figure 5 is to highlight and explain this issue. We have also addressed it in the summary.

In the sensitivity experiments in Section 3.2.3, the authors pointed out the severe dependency on the assumption of cloud fraction and sub-grid decomposition, as well as the timestep. It is generally recognized that it is not straightforward to evaluate the cloud properties of GCMs. The authors should summarize which properties can be evaluated, and the others are more dependent on the model assumptions.

>> We agree that it is not straightforward to evaluate GCMs and observations. We have discussed this when the cloud fraction is discussed in Section 3.1.3 including how it may impact the comparison with observations.

This type of caution should be emphasized in the introduction and the discussion sections.

>> We have modified the text in the conclusions where these comparisons are summarized to emphasize this point (also see below).

The authors may also note differences in the methodology from the cloud evaluation of global storm-resolving models.

>> We have now done this at the end of the conclusions when we highlight the complexities of sub-grid representations, and that these are eliminated when storm resolving models are used.

Specific comments :

p. 3, L 58: The type of the radar (W-band) and the frequency must be indicated in this section.

>> Done

p. 4, Section 2.2.2 and 2.2.3: Specify the method or the definition of the cloud fraction.

>> This is specified in Section 2.2.1 and is the same for both microphysics schemes (clarified)

What are the assumptions of the size distribution? I suggest the full details of the assumptions of the size distributions of all the hydrometeors should be tabulated. The results of the simulator severely depend on them.

>> We have described in more detail the assumptions of the size distributions. Everything is consistent with Fielding and Janisková, 2020 for the single moment simulations, and for the two-moment simulations the size distributions in Fielding and Janisková, 2020 are replaced with the ones described in Morrison and Gettelman, 2008.

p. 5, Section 2.2.4: What is the methodology of the treatment of the sub-columns? Some are described in a later section. This section is the right place to be fully explained.

>> The sub-column approach in the radar simulator (double column approach as defined in Fielding and Janisková, 2020) is described specifically here.

p. 5, L 128-131: As pointed out in General comments, the result from the attenuation calculation of the radar simulator is not understandable. The authors should quantitatively show the quality of the radar simulator in terms of the attenuation.

>> As noted above, there are several sources of loss of the radar signal. Attenuation is only one of them. We have clarified this in the discussion of the simulator, in the results section on reflectivity, and in the summary and conclusions.

p. 6, L153, "the formation of layers with zero fall velocity": I do not understand what "zero fall velocity" means. Are they cloud water or cloud ice, which possesses zero fall velocity?

>> Clarified

p. 6, Figure 1: It isn't easy to distinguish the differences among the results from this figure. The authors could consider a clearer method to show the difference. A possible improvement is to show the difference from the observation for the two numerical results in separate figures.

>> Differences are not easy to distinguish because they are small. We originally made a plot of differences, but since there is still some random noise in the meteorology, it is the average difference that is really important. We now note this explicitly in the text.

p. 11, L251, "The median values of MG3 LWP are comparable to those for the IFS and the observational datasets.": The median value of MG3 LWP is 220, and that for the IFS is 69.8. They are not comparable.

>> The reviewer mis-states the values. The mean (not median) MG3 LWP is 220 g/m2. The median is 101.4 g/m2, 'comparable' to the median observed value of 90.5. The statement has been modified to note it is higher than the IFS median of 69.8.

p. 11, L267, "a greater occurrence of low reflectivity in the upper troposphere related to ice cloud": The radar signal is highly attenuated for the upper troposphere. How reliable are the upper tropospheric clouds?

>> We have discussed this point more fully now, and highlight the issues with the Minimum Detectable Signal. We also note that we do not expect the reflectivity from a ground based radar to be effective at fully characterizing upper tropospheric clouds. The TOA radiative fluxes are better for this.

p. 11, L268, "reflectivity-height histogram": It is generally called a Contoured Frequency Altitude Diagram (CFAD).

>> Noted

p. 11, L272 and after, "left" and "right": These should be smaller and larger values, respectively.

>> We have modified the text to make sure we focus on smaller and larger throughout this section.

p. 11, L274-275, "The model simulations contain a large occurrence of hydrometeors above 5 km that the surface radar would be unable to detect.": Why so many smaller values of radar reflectivity are detected above 5 km for MG3 and IFS (Fig. 5)? This result suggests that the simulator's calculation of attenuation is not adequate.

>> As noted above, this is not an attenuation issue, it is the minimum detectable signal from the radar that falls off with height. This will vary by radar and so it is not part of the radar simulator 'attenuation'. We have clarified this in the text.

p. 11, L280, "This suggests that the MG3 is likely better representing this high-altitude cloud.": The high clouds are severely affected by the model tuning. In which aspect the high cloud is better represented?

>> We now note explicitly what aspects of high cloud are better represented: MG3 allows for an evolution of the size distribution of ice crystals, which better represents variable sedimentation and precipitation processes for ice.

p. 14, L285, "affected by the model sub-grid representation": It must also be affected by attenuation, I guess.

>> Yes, noted in this sentence.

p. 14, L285-307: The description of this part illustrates the difficulty in assessing the nature of GCM clouds. When evaluating them by comparison with observations, the cloud properties in the GCMs are greatly influenced by the assumptions used in the GCMs, and thus, the assumptions themselves are evaluated in some cases. This problem illustrates the difficulty of this study and could be emphasized more. The definition of cloud cover and the subgrid model assumptions are important elements of this uncertainty. In the introduction and the discussion section, this issue, in particular, must be highlighted.

>> Good idea. Added a mention of this issue to the introduction and added a paragraph to the discussion section as suggested.

p. 14, L314-315, "the analysis shows the total low cloud cover in the simulations is reasonable": Please show this result.

>> Changed to say 'radiative effect of clouds' with a reference to Figure 2 (that is really what is important from a climate perspective as noted in the introduction)

p. 15, L336-337, "There is overall more supercooled liquid water": Please indicate where is the location of the 0-degree altitude.

>> Sea surface temperature is about 5°C and the average freezing level is about 900hPa or 0.8km (noted)

p. 15, L339-340, "A common problem is a sensitivity of microphysical processes to the length of the timestep": This serious problem is pointed out abruptly. Do the authors summarize the artificial factors that severely affect cloud characteristics? I speculate that the IFS is not so much affected by the timestep.

>> We have added two references that discuss this sensitivity. A few lines latter we mention that the IFS uses an implicit formulation of the physics which reduces the timestep sensitivity.

p. 15, L349, "supercooled liquid water (SLW)": SLW is already defined at L337. It does not need to be spelled out here.

>> Corrected.

p. 17, L353, "by regime": What is the regime?

>> Noted (low clouds)

p. 18, L377, "a deeper melting layer": Where is the melting layer? How deep is the melting layer of the MG3 scheme?

>> The melting layer is simply the layer in which snow is melting. Not needed here (stated that melting is more active near the surface), so removed this phrase.

p. 21, L425, "the MG3 CAM6 variable fall speed formulation": What is the detail of the formulation of the fall speed?

>> Fall speed is a function of ice and snow size in the default MG3 CAM6 scheme. Noted now.

p. 25, before the end of Section 3.2.3: A lot of the sensitivity results are shown. The authors should summarize the sensitivity experiments by itemizing the main results or in a table quantitatively.

>> Add a summary of the sensitivity tests in a paragraph at the end of section 3

p. 27, L515, "above": Is this higher in altitude or larger in signal?

>> Larger in signal (higher reflectivity). Corrected.

p. 27, L517, "While some of this difference is expected due to attenuation of the observed radar reflectivity that results from rain on the radar radome": This caution must be explicitly described in the main text.

>> Added to section 3.1.3 when radar observations are discussed.

p. 27, L523-329: This paragraph explains the usefulness of the single-column model. I agree. However, I suspect that there is a limitation or caution if it is applied to mid and high-latitude areas.

>> Added a caution due to advection of hydrometeors, but note the timescale for microphysical processes is shorter than an advection time scale, and that this is borne out in the good reproduction of observed cloudiness (fig 4) and LWP (figure 1).

p. 28, L537-539, "The sensitivity simulations show that differences in the autoconversion to rain and accretion onto rain loss processes for liquid water at sub-freezing temperatures are the primary cause of these differences, and the MG3 scheme could be adjusted to enhance the liquid loss process and reduce the LWP in deeper clouds.": If the authors claim that this part is the main conclusion, it should be more emphasized when summarizing the sensitivity results in Section 3.

>> Added this to the summary of the sensitivity tests in section 3

**Review #2**

General Reply: We thank the reviewer for their careful read of the manuscript. The suggestions and clarifications have greatly improved the manuscript. We have made the minor changes requested.

Review of "The Impact of Cloud Microphysics and Ice Nucleation on Southern Ocean Clouds Assessed with Single Column Modeling and Instrument Simulators"

Using an ECMWF SCM, the paper compares two different cloud microphysical schemes, MG (double moment) and IFS (one moment), to ground and satellite based observations over a 2.5 month period (January 1 - March 17, 2017) during MICRE. Large differences of water mass and reflectivity between the schemes are found. Many sensitivity experiments on the cloud microphysics schemes found that Surface radiative fluxes and total water path are highly sensitive to the formation and fall speed of precipitation. Although MG produces well the observed reflectivity and radiative fluxes, higher liquid water contents is seriously overestimated. The sensitivities of the results to liquid Cloud Condensation Nuclei (CCN) and Ice Nuclei are also investigated.

The paper is well written and the results are interesting and important for model improvement.

Recommendations
Minor revisions
Minor points

1) The overestimation of the LWP by MG is too large. EXP Ac3Au1.5 produces less LWP, but still about two times of IFS. Since the single moment can be treated as a special case of double-moment, could an experiment be designed to produce the similar LWP of IFS microphysics?

>> It's more like 50% larger than IFS when autoconversion and accretion are adjusted (Figure 13). It is possible that with further experimentation (fixed number concentration, and low number or large size) the LWP could be further reduced in MG3 to match the IFS. Now noted in section 3.

2) EXP "No Graupel " produces large precipitation, total Ice, and even the liquid is not small, where are the sources of those hydrometers?

>> This is noted in the text and now summarized at the end of section 3. There is a change in the process rates between liquid and ice partitioning that results in more mass of snow (included in IWP).

**Review #3**

General Reply: We thank the reviewer for their careful read of the manuscript. The suggestions and clarifications have greatly improved the manuscript. In response to these points, we have made several major changes. Every figure has been redrafted in response to the queries and concerns to fix units and axes and add panel labels. We have also worked again on the summary and conclusions, highlighting at the end of the text better the major points, and separating a summary from conclusions to highlight the key results and implications for 3D models as suggested. We have answered all the detailed comments and made all the technical corrections suggested (including to all the figures, and reducing the number of figures by combining panels).

Review of "The Impact of Cloud Microphysics and Ice Nucleation
on Southern Ocean Clouds Assessed with Single Column Modeling
and Instrument Simulators"

by Andrew Gettelman, Richard Forbes, Roger Marchand, Chih-Chieh Chen, and Mark Fielding
MS No.: egusphere-2024-599

Recommendations: Major Revision

1 General Comments

This paper describes an experiment comparing two microphysics schemes of differing complexity by testing in a constrained single-column model framework. The results were compared against field observations over an extended period of 75 days. The observations were made in the Southern Ocean where mixed phase cloud processes are important feature of the atmospheric system, and one which is currently poorly represented in numerical weather prediction models. As such this paper makes a useful contribution to the field by evaluating how both a single-moment (1M) and double-moment (2M) microphysics scheme performs in this region. The work takes a broad overview across the full time range, giving an insight into the

problem, whilst also looking in detail at the diurnal cycle. Detailed study of various process rates in the 1M and 2M schemes are trialled and compared.

The observations data come from a 75 day period in Austral spring, on a remote Island in the Southern Ocean. Instrument simulators are employed to convert the model predictions into a form that can be directly compared to observations, e.g. hydrometeors are passed through a radar simulator code.

The major conclusions are that both the 1M and 2M schemes have some skill in representing the observed cloud and rainfall properties, both at the surface, top of atmosphere, in in the vertical cloud column, especially for bulk properties such as mass concentrations and liquid water paths (LWPs), including supercooled liquid water (SLW). The 2M scheme is shown to add value for radar reflectivities - hence clouds structures, but at the expense of somewhat overestimated LWP.

The results of a suite of tests are then presented, including time step sensitivity, process rate comparisons between schemes, and numerical formulations of some aspects of the microphysics - namely sedimentation, along with sensitivity tests of number concentrations of cloud condensation nuclei (CCN) and ice nucleating particles (INP).

The paper could be improved by adding clarity and perhaps a re-formulation of the aims and conclusions of the paper. It is not clear who the intended audience is, either Southern Ocean specialist, cloud model developers or modelling centres (numerical weather prediction (NWP) or climate) wanting guidance when choosing between 1M and 2M cloud microphysics. It might be all three but, some rewriting to tighten up the aims and conclusions is suggested.

For example - Line 71 suggests there will be guidance on where 2M schemes may be necessary - but this isn't directly concluded upon later.

>> This is now commented on more explicitly in the discussion and conclusions, in response to previous comments.

it might be the case (as is in fact stated by the authors) that full 3D simulations are required to go beyond the results in this work, something which would clearly require a step-change in compute resource. As well as highlighting that changing various parameters and assumption(e.g. number concentrations of aerosols) can alter the results, the work would benefit from a deeper commentary looking at the relative importance of the various aspects tested. otherwise this paper reads like a description of a test framework, as a forerunner to full 3D model simulations.

>> We have added some commentary on this also in response to other comments.

>> More substantially, we have gone over the introduction and conclusions again with an eye towards tightening up the aims of the paper and to make them more consistent.

2 Specific Comments

2.1 Title

Mention of the comparison of 1M and 2M microphysics schemes would be more informative of the content of the paper. Also mentioning that the key difference is enhanced SLW in the MG3 2M scheme would point to one of the key results.

>> Noted in the introduction and in the conclusions now.

2.2 Abstract

it is stated that both dynamic and aerosol through microphysics are important, but the model tests only use single-column, and so it isn't clear what the "dynamic" refers to here - is it just the tests where the relaxation to prescribed temperature was turned off?

>> Yes. Noted in the text now in the conclusions with an extra phrase in the discussion of the SCM configuration.

2.3 results

As a general comment - can the number of figures be reduced? For example, as mentioned elsewhere - can Fig. 6 be combined with 11 and 12 as anomalies?

>> We originally had these figures combined, and it was a mess. The fewer number of lines on a plot is necessary for the figures to be read appropriately.

Are all process rate figures needed (Figs. 8, 9, 10, 12)- they are also difficult to read. perhaps there could be a cloud liquid ice 2x2 and a recipitation rain and snow 2x2 panel?

>> We have combined the process rate figures as suggested into a liquid-ice and then a rain-snow figure. Thank you for this suggestion.

– Lines 186-198 are repetition with the figure caption, read like methodology, and could possibly be tidied up or consolidated.

>> Consolidated (removed some of the descriptive text to the caption, deleted repetitive text).

– Lines 212-216 - Stated that there is too much high cloud in IFS, resulting in poor TOA LWperformance. this is followed up on in Sect. 3.1.3. it is mentioned that there are Line 277 states that IFS produces more of the low reflectivity values at altitude than MG3. But IFS also

has a shoulder of higher reflectivity values (Fig. 5 middle right) compared to MG3 - could these values not be responsible for the poor TOA LW performance?

>> Yes. Modified the text around line 277

– Lines 229 onwards - There is discussion around the limitation of CERES SYN at night - should this data even be shown (possibly yes, to highlight the issues)?

>> Yes it should be shown, as the reviewer notes, to highlight the issues.

– The use of MODIS at high latitude is interesting, and shows some evidence of a diurnal cycle - soon after sunrise - is this also an artefact, or believable? There is not much discussion on this particular aspect.

>> There is a whole paragraph devoted to discussion of these artifacts in the 'CERES-SYN' data (based on MODIS) in section 3.1.2 (2nd paragraph), and an extra daytime MODIS data set added to Figure 2 to highlight this.

– Line 334 - A paragraph here to introduce the sensitivity tests in Sect. 3.2 would aid clarity of the following sections.

>> Added a little more guidance and introduction to the sensitivity tests as suggested.

– Line 405 - the timesteps tests for IFS seem unnecessary.

>> It is stated in the text that there is little sensitivity, but it's easy enough to show the lack of sensitivity explicitly.

– Figures 7, 8, 9, 10 - can resultant budgets be included here? For example Line 358 onwards, discussion of Bergeron process indicates that some processes not fully compensated for between IFS and MC3 - this could be shown visually too.

>> We don't really have a full set of budget terms. They could be constructed from the process rates, but they become a bit less informative for comparison between the schemes and would add more figures to the text.

– Section 3.2.2 - authors comment that there aren't huge differences in outcomes, perhaps only that MG3 has more SLW. Does this section add a huge amount to the overall paper?

>> The SLW differences and the differences in precipitation summarized at the end of this section we think are actually pretty relevant, as well as the fact that compensation exists. There are not many studies that are able to do such a detailed analysis of two very different microphysical schemes in a large scale model, so we think this is important to leave in. We note now at the top of section 3.2 why we are putting this section in, and have commentary to the

conclusions discussing the results of this section. We think this actually improves the paper and the conclusions, so we would like to keep this section.

– There are significant differences in model results when scaling either CCN or INP by factors of 4, as might be expected, but it's not clear if these values a physical / climatologically reasonable or just for illustration.

>> It is for illustration, but CCN can easily vary by an order of magnitude over the S. Ocean. Added a reference to observations from recent field campaigns adjacent to MICRE that show this.

Do these test point to the need for use of 2M in NWP and climate simulations or not? Varying parameters individually will give different results, but the true test is covarying parameters, in a 3D model - which whilst commented on in the summary, does beg the question as to what these tests are trying to illustrate.

>> These tests are trying to illustrate the underlying process differences and where to focus sensitivities. When we get to a 3D simulation, this is a road map for understanding differences in ways that are a lot harder in an interactive simulation where the microphysics feeds back on the thermodynamics. We do spend an entire paragraph in the summary on this point. We have added a concluding sentence that makes the point more definitive.

2.4 Summary and Conclusions

This section would benefit from editing, in particular to link more closely to the abstract and aims. One aspect seems to be the importance of sub-grid representation of precipitation and cloud fraction, which is talked about in the results text. The paper shows that both IFS and MG3 can represent portions of the cloud fields, that surface radiation is OK, and that where data are good the LWP predictions are good, although perhaps MG3 is too large, even though this doesn't feedback on to SW biases at the surface.

>> We have gone over the summary again and tightened it up. We comment a bit more on attenuation, and have rewritten the last few paragraphs to be shorter and focus more on key messages and future work with 3D models. It's about 10 lines shorter now.

3 Technical Corrections

– Line 205 - "TOA flux" should be "TOA SW Flux"

>> Changed

– Units should be represented as negative powers where appropriate and have a space between elements

>> Changed in all the figures (this required remaking all of them)

– Line 360 - incorrect abbreviation of 'accretion' (also elsewhere, plus others not defined)

>> Fixed. Only processes discussed are defined for clarity.

3.1 Figure comments

3.1.1 General

There are numerous formatting styles in use for figures in this manuscript and general inconsistencies that could be rectified.

– Many multiple panel plots need sub-panel labels, (a), (b), (c), etc, and the text updating accordingly

>> Done

– Suggest that panel titles are removed and moved into the captions

>> We feel titles are needed for clarity

– suggest moving variables names and units from panel titles onto y-axes

>> Done

– units should be written as negative powers as appropriate, and spaces between elements (also in the body text)

>> Done

– units should be enclosed in square brackets consistently

– Figures 6, 11, 12 only have y-axis titles on the top row

>> Fixed

– Figure 2 needs the dashed-line describing in the caption

>> There is no dashed line in figure 2. In figure 1, the dashed line is just the IFS (data sources are now put in the caption by color)

3.2 specific

– Figure 1 (and other time series) - could the time units be in days?

>> Done

– Figure 1 - can the xtick but in 6 hourly (or 4?) intervals for 24 hours?

>> We tried this, but it got a bit too noisy to put that many ticks on the axis.

– Could Figure 6 be combined with Figs. 11 and 12 in to a 5x3 panel plot, and the figures 11 and 12 presented as anomalies, as the results are very difficult to visualise

>>  As noted. We tried combining Figure 11 and 12, and it's really a mess with something like 20 lines. It is much better to have separate figures. We tried re-making figures 11 and 12 as anomaly plots, and these are difficult to interpret because you do not have the mean value to guide you about what is a significant difference or not. So we think sticking with the current figures are better

– Figures 8 through 13 - many of the abbreviations of process rates are not defined, or incorrectly defined (typographical errors). Could a lookup table be provided perhaps?

>> Only Figure 7 through 10 have process rates. Figure 11-13 have sensitivity tests which already are defined in a table. We checked these match the figures.

>> We have corrected any typographical errors. We also added two new tables in an appendix to describe all the process rates (Liquid and Ice, Rain and Snow) and indicate which scheme they are used in. This is now referred to when the process rate figures are mentioned, and in the captions.

---

## Referee Report (RR1)

**Response to Authors Response to Review of "The Impact of Cloud Microphysics and Ice Nucleation on Southern Ocean Clouds Assessed with Single Column Modeling and Instrument Simulators"**

by Andrew Gettelman, Richard Forbes, Roger Marchand, Chih-Chieh Chen, and Mark Fielding

MS No.: egusphere-2024-599

**Recommendations: accepted as is**

**1   General Comments**

The authors have responded to my review and made a number of changes to the manuscript which has improved the quality of the work and satisfied the queries that I had made.

In particular

- the more explicit guidance on 2M schemes which has now been added helps bring out the important conclusions of the work

- additionally commentary on the utility of single column modelling as a tool and forerunner to full 3D simulations has been added.

- the summary and conclusions are now much clearer following a rework which has aided understanding.

- the figures were redrafted and combined where appropriate which has added clarify and readability to the paper.

- the particular importance of supercooled liquid water and the results of process rate analysis now stand out more clearly.

- A table of abbreviations of process rates has been added.

- the significance and importance of varying CCN and IN concentrations is now more explicit following addition to a reference to field studies in the region.

- various typographical errors and minor changes to formatting have improved the manuscript